health and disease and epidemiology/ computational biology

simulation, individual-based model, infectious disease

**Author for correspondence:**
Frank Krauss
e-mail: frank.krauss@durham.ac.uk

# JUNE: open-source individual-based epidemiology simulation

Joseph Aylett-Bullock[1,2,†], Carolina Cuesta-Lazaro[1,3,†], Arnau Quera-Bofarull[1,3,†], Miguel Icaza-Lizaola[1,3,‡], Aidan Sedgewick[1,4,‡], Henry Truong[1,2,‡], Aoife Curran[1,3], Edward Elliott[1,3], Tristan Caulfield[7], Kevin Fong[8,9], Ian Vernon[1,6], Julian Williams[5], Richard Bower[1,3] and Frank Krauss[1,2]

[1]Institute for Data Science, [2]Institute for Particle Physics Phenomenology, [3]Institute for Computational Cosmology, [4]Centre for Extragalactic Astronomy, [5]Institute of Hazard, Risk and Resilience, and [6]Department of Mathematical Sciences, Durham University, Durham DH1 3LE, UK
[7]Department of Computer Science, and [8]Department of Science, Technology, Engineering and Public Policy, University College London, London WC1E 6BT, UK
[9]Department of Anaesthesia, University College London Hospital, London NW1 2BU, UK

 JA-B, 0000-0001-7551-3423; AQ-B, 0000-0001-5055-9863; AS, 0000-0002-9158-750X; HT, 0000-0002-0105-1278; FK, 0000-0001-5043-3099

We introduce JUNE, an open-source framework for the detailed simulation of epidemics on the basis of social interactions in a virtual population constructed from geographically granular census data, reflecting age, sex, ethnicity and socio-economic indicators. Interactions between individuals are modelled in groups of various sizes and properties, such as households, schools and workplaces, and other social activities using social mixing matrices. JUNE provides a suite of flexible parametrizations that describe infectious diseases, how they are transmitted and affect contaminated individuals. In this paper, we apply JUNE to the specific case of modelling the spread of COVID-19 in England. We discuss the quality of initial model outputs which reproduce reported hospital admission and mortality statistics at national and regional levels as well as by age strata.

## 1. Introduction

The spread of SARS-CoV-2 in populations with largely no immunological resistance, and the associated COVID-19 disease,

†Equal contribution.
‡Equal contribution.

have caused considerable disruption to healthcare systems and a large number of fatalities around the globe. The assessment of policy options to mitigate the impact of this and other epidemics on the health of individuals, and the efficiency of healthcare systems, relies on a detailed understanding of the spread of the disease, and requires both short-term operational forecasts and longer-term strategic resource planning.

There are various modelling approaches which aim to provide insights into the spread of an epidemic. They range from analytic models, formulated through differential or difference equations, which reduce numerous aspects of the society–virus–disease interaction onto a small set of parameters, to purely data-driven parametrizations, often based on machine learning, which inherently rely on a probability density that has been fitted to the current and past state of the system in an often untraceable way. As another class of approaches, agent-based models (ABMs) are *particularly useful when it is necessary to model the disease system in a spatially-explicit fashion or when host behavior is complex[.]* [1, p. 2:5].[1] Being the traditional tool of choice to analyse behavioural patterns in society, they find ample use in understanding and modelling the observed spread of infections and in leveraging this for intermediate and long-term forecasting [3–5]. Such models also provide the flexibility to experiment with different policies and practices, founded in realistic changes to the model structure, such as the inclusion of new treatments, changes in social behaviour and restrictions on movement.

To simulate pandemics, specific realizations of ABMs, individual-based models (IBMs), have been developed in the past two decades, for example [6,7]. In these models, the agents represent individuals constituting a population, usually distributed spatially according to the population density and with the demographics—age and sex—taken from census data.[2] Within the existing taxonomy of agent-based models in epidemiology, see for instance [8,9], these models often use a disease-specific modelling framework. Interactions between individuals in predefined social settings, systematically studied for the first time in [10], provide the background for disease spread, formulated in probabilistic language and dependent on the properties of the individuals and the social setting. The sociology of the population and the transmission dynamics are constrained separately using external datasets and available literature, and connected in the description of the spread of the disease. Calibration of such models to observed disease outcomes, such as hospital admission and mortality rates, is therefore reduced to the specific interface between the disease and the varying physiology across the broad population. Policy interactions and mitigation strategies can be flexibly encoded in detail as modifications of the social setting, and allow precise analysis of their efficacy that is not readily available in other approaches.

Evidence from disease data such as COVID-19 fatality statistics suggests that case and infection fatality rates are correlated, amongst other factors, to the age and socio-economics status of the population exposed to the etiological agent [11]. This necessitates the construction of a model with exceptional social and geographic granularity to exploit highly local heterogeneities in the demographic structure. In this publication, we introduce a new individual-based model, JUNE,[3] a generalizable modular framework for simulating the spread of infectious diseases with a fine-grained geographic and demographic resolution and a strong focus on the detailed simulation of policy interventions. JUNE reaches a geographic resolution of societal factors similar to models that focus on single-site infection models, such as [12], where space, location and distance are carefully modelled. In addition, similar to approaches such as the STHAM model [13], the individuals in JUNE follow detailed spatio-temporal activity profiles that are informed by available data including time surveys, geographical and movement data. In contrast to such models that are usually constrained to a few tens of thousands of agents, JUNE simulates, simultaneously, the full population of a country in its spatio-temporal setting, and how a disease spreads through its population mediated by contacts between individuals. JUNE allows for flexible and precise parametrizations of policies that affect groups of individuals selected according to any of their characteristics. This allows modelling of policies to mitigate the further spread of a disease, realized as changes and restrictions on movement, to which we add the effectiveness of changes in social behaviour such as social distancing. The major cost for this level of detail in the model is in

---

[1]Indeed, many models also feature some optimizing behaviour of individuals as artificial intelligence-type actors against randomly drawn welfare functions, e.g. [2].

[2]We will use the term 'sex' in regard to chromosomal differentiation throughout this paper rather than gender. At the time of writing a full classification of the impact of chromosomal sex versus gender identification on the epidemiology of COVID-19 is unavailable. Within our modelling framework, nested and non-nested identifiers can be constructed to map sex and gender should more granular statistical data be available.

[3]A full open source code base and implementation examples are linked here: GitHub: https://github.com/IDAS-Durham/JUNE and PyPI: https://pypi.org/project/june/. The version used for this paper is v. 1.0

computational load; indeed, models such as JUNE would probably not have been possible prior to 2010 without using a prohibitive amount of computing power, see for instance [14].

As a first application of JUNE, we model the spread of COVID-19 in England. In this context, JUNE uses census, household composition and workplace data to ensure that each of the 53 million people in England are assigned a specific, identifiable location at any point in time. Their activities, health, age and other demographic attributes are then modelled at a fine-grained geographical level, which helps to ensure that local heterogeneity in population and movement characteristics are well recovered. This societal structure, generated by the model, is validated against a series of datasets (among others this includes: surveys of household size and composition, location and size of businesses, size and type of schools by region). The calibration to observed data from the actual spread of SARS-CoV-2 is then limited to how the virus is transmitted in the community through person-to-person 'contacts' (in the sense of sufficient proximity and timing to transmit). This component of the infection is calibrated to the spatio-temporal development of hospitalizations and casualties during the COVID–19 outbreak in England, starting in early March 2020. Preliminary observations demonstrate that a detailed large-scale model of this type has important implications for intermediate- to long-term modelling of the SARS-CoV-2 spread in the UK and elsewhere.

The remainder of this paper is as follows. Section 2 provides an overview of the structure of the JUNE framework. In §3, we detail the construction of a virtual population including a variety of demographic attributes. For the example case of England, we demonstrate that the constructed population reproduces the distributions of age, gender, ethnicity, socio-economic indices and the composition of the households they live in, all with a granularity of a few hundred people. The static properties of the population also include the assignment of students to schools and universities and of employment in companies dis-aggregated by 21 industry sectors. In §4, we discuss the dynamics of the population model. We demonstrate how JUNE correctly reproduces the average time-profile of daily activities of individuals in England. We also describe in detail how we reconstruct movement and daily commute patterns based on publicly available data. Social interactions in various settings are modelled through parameters informed by social mixing matrices derived from surveys such as `PolyMod` [10] and the BBC Pandemic project [15]. In contrast to other models, JUNE also incorporates interactions in various social venues such as pubs, restaurants, cinemas and shopping, outside the more structured settings of households, workplaces and schools. Section 5 introduces the generalizable disease model with specific applications to COVID-19—its transmission properties and the impact it has on infected individuals. We employ a probabilistic model for the former, while for the latter we incorporate data from the UK and other countries to characterize the journey of infected people through the healthcare system. In §6, we describe how JUNE models the impact of various policy interventions and other mitigation strategies. In §7, we show some first indicative results of JUNE highlighting its potential for future, more detailed studies. Section 8 introduces our approach to fitting the model using Bayesian emulation. We summarize our work in §9, and conclude the paper with discussion of future work and improvements to the model.

# 2. The structure of the JUNE modelling framework

The JUNE framework is built on four interconnected layers: `population`, `interactions`, `disease` and `policy`, the layers and their interfaces are illustrated in figure 1. In the context of this publication, we focus on the application of JUNE to England's population, the spread of the COVID-19 disease, and policies that have been enacted by the UK Government in 2020. Clearly, a different population with different behavioural patterns will not only affect the distribution of individuals according to their personal characteristics, but it will also necessitate the adaptation of e.g. social venues to these patterns and corresponding changes to the `population` and `interactions` layers. Similarly, modifications to the `disease` layer will allow application of the JUNE framework for a different disease or, possibly, even a range of competing diseases. This flexibility and adaptability is even more pronounced in the `policy` layer where the introduction of new policies in reaction to an epidemic depends on behavioural patterns or societal norms.

The `population` layer encodes the individuals in the model and constructs static social environments such as the households they live in, the schools and universities they study in, and the workplaces where they work. The construction of the virtual population is informed by demographic data such as age, sex and ethnicity distributions, the geographical location of their residence, and its composition. Depending on their age, individuals will attend school or university, work, or be retired.

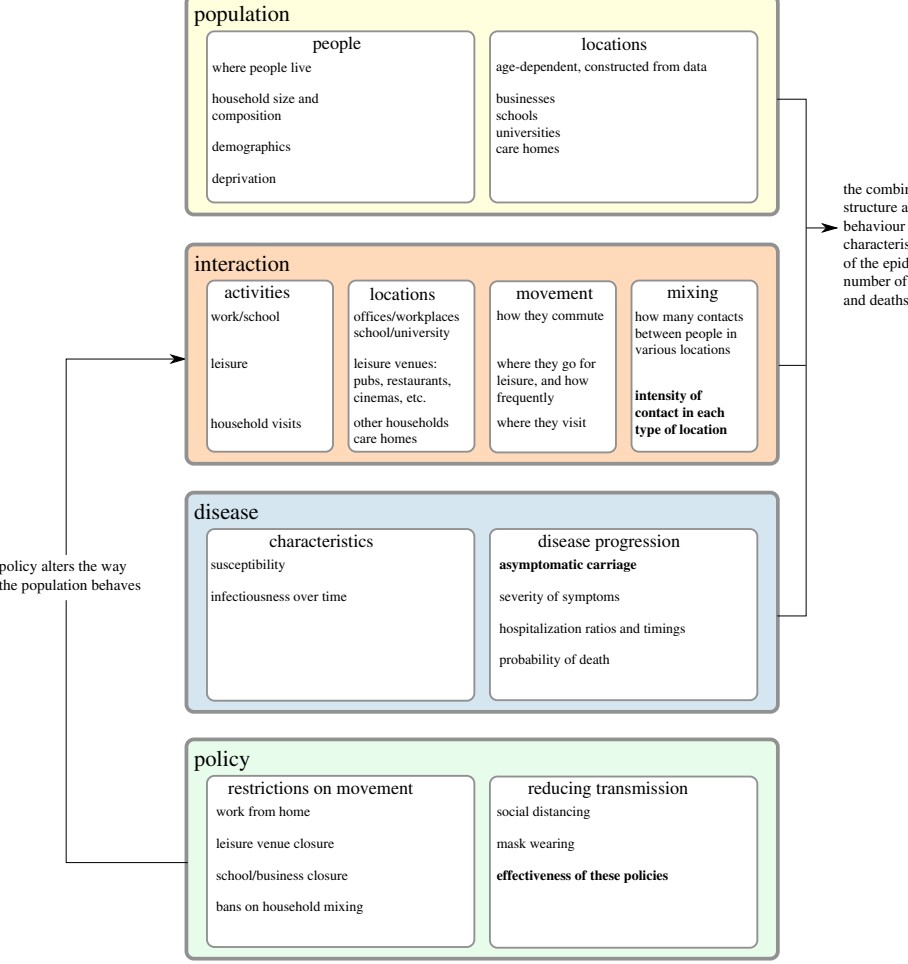

**Figure 1.** Overview of the structure of JUNE. Fitted parameters are shown in bold.

The interaction layer models the social interactions of individuals, based on data about the frequency and intensity of contacts with other people in social settings. In addition to daily patterns of regular interactions with fixed groups of individuals such as household members, students and teachers in schools, and work colleagues, the interaction layer also models more randomized interactions. These include daily commute patterns to and from work, and more dynamic activities such as visits to restaurants, pubs, cinemas and visits to other households.

The disease layer, which sits on top of the population and interaction layers, models the characteristics of disease transmission and the effects it has on those infected. In terms of disease transmission, the model incorporates the varying susceptibility of individuals, how likely individuals are to become infected when they mix with others in various locations, and how infectious they are over the course of their infection. In terms of disease progression, the model captures how likely individuals are to experience symptoms with varying severity, to be hospitalized, to be admitted to intensive care, or to die, as well as the timings associated with these events.

In response to the spread of a disease through its population, a government might introduce policy measures designed to control and reduce the impact of the disease. In the case of COVID-19 in England and many other countries, policies have included social distancing measures, the closure of schools, shops, restaurants and other leisure venues, and restrictions on movement. In JUNE, these are modelled in the policy layer. The high level of detail present in the population and interaction layers allows policies to be modelled at a corresponding granularity. This enables JUNE to describe the impact of policies that can be applied to specific geographical regions, to specific venues or sectors, or to individuals with specific characteristics. Examples include, but are not restricted to, the closure of targeted different types of (or even singular) venues, the inclusion or exclusion of specific age groups when going to school, shielding of the older parts of the population, modification to inter-household visits, and self-isolation measures for infected individuals and their contacts, including variations of compliance with these measures.

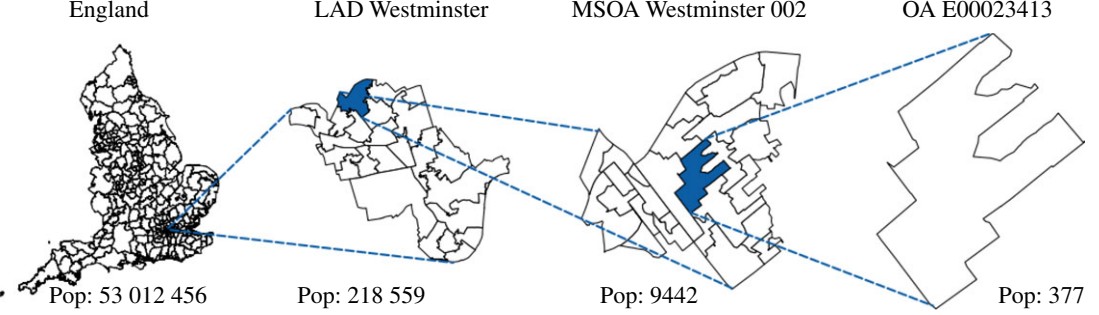

| England | LAD Westminster | MSOA Westminster 002 | OA E00023413 |
|---|---|---|---|
| Pop: 53 012 456 | Pop: 218 559 | Pop: 9442 | Pop: 377 |

**Figure 2.** Graphical representation of how the census data for England are structured, from the level of local authority districts (LAD), down to the level of output areas (OA), with middle layer super output area (MSOA) in between.

# 3. Population and its static properties

June creates a detailed virtual population at the individual level through its `population` layer, using a cross-section of demographic and geographic information. Since June relies on multiple datasets, and is built to dynamically adapt to varying types of input, the approaches described in this section are generalizable to other settings with similar or complementary data availability. Given that different settings, e.g. countries, may have different methods and types of data collection, many of the input parameters described here are optional, allowing June to be more easily adapted to differences in reporting.

## 3.1. Geography and demography

To facilitate generalizability across multiple settings, June models the geographic distribution of a population using a hierarchy of three layers—regions, super areas and areas. Layering these geographies allows the use of data at different levels of aggregation and enables simple statistical projections of data between these levels.

For the case of England, the construction of the virtual population in June is largely based on data from the latest UK census, which was carried out in 2011. This data is accessible through NOMIS, an open-access database provided by the Office for National Statistics (ONS), and each dataset varies in its degree of aggregation. The three hierarchical geographical layers represented in figure 2 are:

1. regions—London, East Midlands, West Midlands, the Northwest, the Northeast, etc.;
2. super areas—approximately 7200 middle layer super output areas (MSOAs);
3. areas—approximately 180 000 output areas (OAs).

The individuals in June's virtual population are constructed according to age and sex disaggregated information, the minimal information required by June. In the case of England, the ONS census data provides this information at the OA (area) level [16,17] such that June naturally captures the population density at the most fine-grained level. In figure 3, we show age distributions in different regions. We use data derived from the ONS to additionally assign one of five broad ethnic categories to individuals based on their age, sex and location of residence [18] and follow a similar procedure for the socio-economic index, which we divide into centiles, according to the ranked English Index of Multiple Deprivation (IMD) [19].

## 3.2. Household construction

The virtual population within June is placed into households of varying types. Depending on the structure of the available public records, households in June can be allocated with an arbitrary degree of granularity, taking into account multiple demographic attributes.

For the UK the ONS census datasets provide a detailed record of both household type and composition in England at the OA (area) level. That is, for each OA there is a set of summary statistics across a number of criteria, choices can then be made in regard to aggregating those frequency measurements at different resolutions. In term of data categories for households, the OA (area) level provides the following occupancy type counts: single, couple, family, student, communal and other

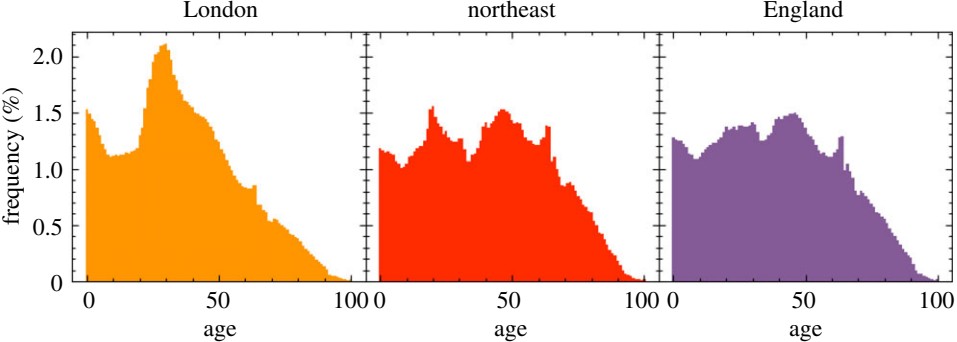

**Figure 3.** Age profiles in different regions of England, taken from the ONS database and implemented in June.

[20], and further specifies them by the number of old adults, aged over 65, adults, dependent adults (such as students) and children, providing around 20 distinct classes contingent on the underlying census information. Given the data structure, it is impossible to recover the exact composition for each household type. For example, the number of non-dependent children (people over the age of 18 living with their parents), the number of multi-generational families, and the exact distribution of adult groups sharing a household are not specified in these datasets. However, these features can be statistically extrapolated using a mix of further secondary data and validated against various aggregate survey information at the regional and national level.[4] Households are populated iteratively giving preference to those household types with the most precise available data. The exact procedure for the UK is documented in appendix A.1.

Similarly, to households, care homes are classified by type, positioned and populated using ONS data [21] at the OA (area) level. The ONS collects information on the age distribution and sex of residents of communal establishments at the MSOA (super area) level [22]. By combining these datasets, we infer the age and sex distribution of the care home population.

Other communal establishments specified in the census, including student accommodations and prisons can be flexibly added with sufficient datasets. Within the presented version of June we do not model explicitly the age and sex distribution of these other communal establishments, however, since the age and sex distribution of the OA (area) level's population will be biased towards these communal residents, their resident characteristics are deemed to be realistic. Cross-checking of case studies suggests that the communal allocation does capture the age and sex very accurately.

## 3.3. Construction of virtual schools and universities

Schools and universities are two locations where a resident population will visit and interact. Every location can have universal and specific attributes flexibly initiated within the modelling framework depending on the detail of available information. From public records June locates and enrols schools according to their precise geo-coordinates and the publicly reported age ranges and numbers in attendance at each school. Students are sent to one of the $n$ nearest schools to their place of residence, according to which schools cater for their age. We form year groups which include all students of the same age. The formation of year groups, and classes within them, allows June to control mixing within and between children of different ages within the school environment.

To model schools in England, we use data provided by [23] to determine the location of schools and their age brackets. Based on the current enrolment requirements for the UK, we assume that children between the ages 0–19 can attend school, with mandatory attendance between 5 and 18. Since 19-year-olds can attend school, university, work or none of these, the institution they attend is determined by the number of vacancies in schools accepting students of that age group. We send children to one of the $n = 10$ nearest schools where classes sizes are limited to 40. One way in which we validate our assumptions is by comparing average travel distance to schools of different types. In June, we find 1.7 and 5.0 km for primary and secondary school students, compared with 2.6 and 5.5 km, respectively, from the 2014 national travel survey [24]. Teachers are allocated to a school by randomized sampling from the available population—i.e. people over the age of 21 (to allow them to

---

[4]A forthcoming publication will discuss the use of secondary data to further constrain the uncertainty in the household construction and the subsequent impact on simulating the spread of COVID–19.

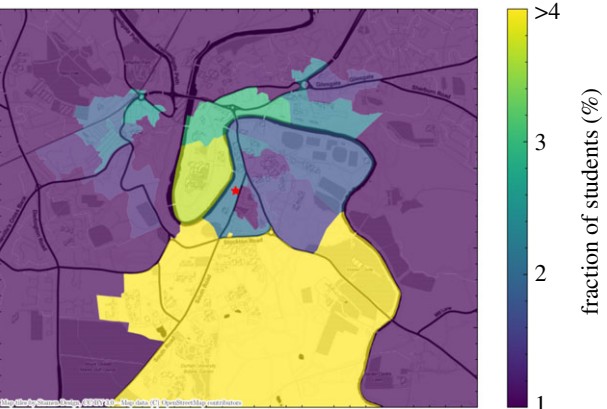

**Figure 4.** A geographical visualization of the location of student residences in Durham in J<small>UNE</small>, with the university location represented as a red star in the middle. Output areas are colour-coded according to the fraction of students they host. Note that the large southern area is where most of the university accommodation blocks are located.

have attended university), who live in the same MSOA (super area) as the school, and who have been assigned 'Education' as their work sector (see next section for more details on work sectors). The number of teachers assigned to a particular school, and therefore the number of classes, is determined by sampling the ratio of students to teachers from a Poisson distribution with mean equal to the UK national average, separately for primary (mean of 21) and secondary schools (mean of 16), or a random choice of the two for mixed schools [25]. J<small>UNE</small>'s recovered student–teacher ratios are 22.0 and 17.8 for primary and secondary schools, respectively.

Similarly, universities are located according to their address as recorded in the UK Register of Learning Providers (UKRLP) [23]. Students are enrolled in a university using the UKRLP enrolment data. The enrolled students are assigned from a subset of the local population to the university, reflecting the fact that the ONS census uses the term-time address of students.

Students are sampled from adults between the ages 18–25 with a preference given to those previously assigned to living in student or communal households in a given radius around the university. The concentrations of students expected by J<small>UNE</small> in a particular area can be matched to secondary data on student living within any given city in the UK. Figure 4 shows an example of a university city, Durham, in which we highlight the modelled regions inhabited by students. To date, we have not explicitly constructed the employees at universities and their interactions with the student body.

## 3.4. Construction of workplaces

Workplaces are constructed for the subset of the population in employment according to public records. We divide employment structures into three categories: work in companies with employees; work outside fixed company structures; work in hospitals and schools. The number of employees in each MSOA (super area) is data driven from the workforce information in that specific MSOA. To distribute the workforce over workplaces, J<small>UNE</small> first initializes companies based on data containing their locations, sizes and the sectors in which they operate. In a next step, individuals who are eligible to work (i.e. between the ages of 18–65) are assigned an industry sector based on the geographical distribution of where the workforce live by sector. This results in origin–destination matrices which are used to match workers to their workplace and to optimize the distribution of individual company to reproduce sector-dependent distributions.

In England, the ONS database contains information on companies and workforce structured by industry type. Industries and companies are categorized according to 21 sectors following the Standard Industrial Classification (SIC) code convention [26] (table 1) and information about company numbers per sector, and company sizes is available at the MSOA (super area) level [27]. Similarly, the ONS data also contain the size and sex distribution of the workforce by sector at the MSOA level, as well as the location of their employment [28,29]. This enables the construction of an origin–destination matrix and allows us to distribute the workforce accordingly. More details on this specific procedure for initializing companies in J<small>UNE</small> and matching working individuals to these companies can be found in appendix A.3.

**Table 1.** Standard Industrial Classification (SIC) code identifiers for the 21 workplace sectors modelled in June and used by the ONS to categorize companies [26].

| SIC code identifier | description |
| --- | --- |
| A | agriculture, forestry and fishing |
| B | mining and quarrying |
| C | manufacturing |
| D | electricity, gas, steam and air conditioning supply |
| E | water supply; sewerage, waste management and remediation activities |
| F | construction |
| G | wholesale and retail trade; repair of motor vehicles and motorcycles |
| H | transportation and storage |
| I | accommodation and food service activities |
| J | information and communication |
| K | financial and insurance activities |
| L | real estate activities |
| M | professional, scientific and technical activities |
| N | administrative and support service activities |
| O | public administration and defence; compulsory social security |
| P | education |
| Q | human health and social work activities |
| R | arts, entertainment and recreation |
| S | other service activities |
| T | activities of households as employers; undifferentiated goods-and services-producing activities of households for own use |
| U | activities of extraterritorial organizations and bodies |

The resulting distribution of our procedure assigning individuals an industry sector can be seen in figure 5. June captures many of the sex-dependent features of the job market such as females dominating the healthcare profession and males the manufacturing sector. Recovering these sector-level sex imbalances can be crucial to reproducing and predicting potential sex imbalances in disease spread.

June locates employment using data specifying the physical position of, for instance, company buildings. This, however, does not capture other modes of employment. We model people working from home through the specification of single-person companies in the same location as their place of residence. It should be noted that we do not currently explicitly model those workers who may not work in formal company buildings but also do not work from home, such as contractors who may interact with a household of the people they are visiting for building improvements or maintenance work.

Hospitals play a dual role in June, both as an essential part of patient's possible medical journey and as workplaces. We will discuss the role of hospitals for the former case in §5. For both purposes, hospitals are initialized like many other locations in June, based on available data regarding their location and capacity. Hospitals can be modelled individually or as clusters; in the latter case we represent the full cluster by one hospital. For our simulation of COVID–19 in England, we define the relevant National Health Service (NHS) trusts as those that reported disease-related casualties—this amounts to a total of 129 trusts—and we cluster them into single hospitals.[5] The clustering of hospitals is in fact a better representation of the situation in England. The aggregation of data by NHS trust allows for a more detailed comparison of the number and geographical spread of hospital admissions with available data. We assign medical workers to hospitals based on the same origin–destination matrix at the MSOA (super area) level as derived above, by choosing from those who work in the healthcare

---

[5]Some NHS trusts share resources and exchange patients across regions in an ad hoc manner; however, this is not modelled explicitly.

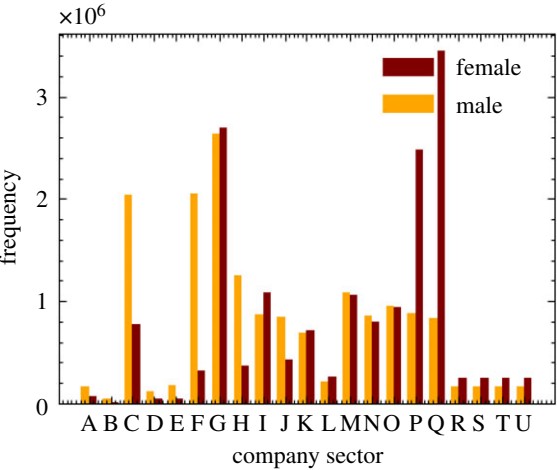

**Figure 5.** Number of workers by sex and company sector (denoted by SIC code identifiers, see table 1) in JUNE.

sector (Q), with the additional constraint of assuming a fixed ratio of 10 hospital beds per medic–nurse or doctor. Teachers are chosen from the population in a similar matter by using the origin–destination matrix and choosing from those in the education sector (P).

# 4. Simulating social interactions

The `interation` layer maps the spatial movement, location and intensity of social interactions, of the virtual population. To ensure a close match to real-world dynamics, summary information on the virtual population is calibrated to equivalent observed summary data. Comparable IBMs, such as [6,7], simulate social interactions in either static environments, such as households, schools, or workplaces, in a similar manner to that described in the previous section, or in a less specific way determined by gravity models. By contrast, JUNE allows for the specification of additional social settings, and directly connects them to geographical locations, such as shops or restaurants. We can also model transport routes of different types between specified geographical start- and end-points. This granularity is further increased through the addition of social mixing matrices which parametrize differences in frequency and intensity of contacts between individuals in various settings [10,30].

## 4.1. A virtual individual's day

Calendar days, decomposed into time-steps of varying length given in units of hours, are the background for our simulation of the social interactions of our virtual population.

Time in JUNE occurs in discrete time-steps of varying length measured in hours. Every time-step in JUNE is tagged to a calendar day. The use of calendar time allows JUNE to distinguish between week-day and weekend activity profiles, which is relevant for time spent at work or in school. Each day can have a number of fixed, static, activities, such as 8 h of work at the workplace or 10 h at home overnight, supplemented with other activities, denoted as 'other', that are distributed dynamically. Time-steps apply to all individuals, and are chosen to best approximate an 'average' individual's day. The default time-steps are described more explicitly in appendix B. During each time-step in which an 'other' activity is allowed, each person who is not otherwise occupied, for example they are working or ill and in hospital, is assigned a set of probabilities for undertaking other activities in the model. These probabilities are part of a flexible social interaction model and depend on the age and sex of the person.[6] Given $N$ possible activities with associated probabilities per hour given by $\lambda_1, \ldots, \lambda_N$, for a person with characteristic properties $\{p\}$, the overall probability $\mathcal{P}$ of being involved with any activity in a given time interval $\Delta t$ is modelled through a Poisson process,

$$\mathcal{P} = 1 - \exp\left(-\sum_{i=1}^{N} \lambda_i(\{p\})\Delta t\right). \tag{4.1}$$

---

[6]These probabilities can be generalized to depend on any attributes of the individual given reliable data.

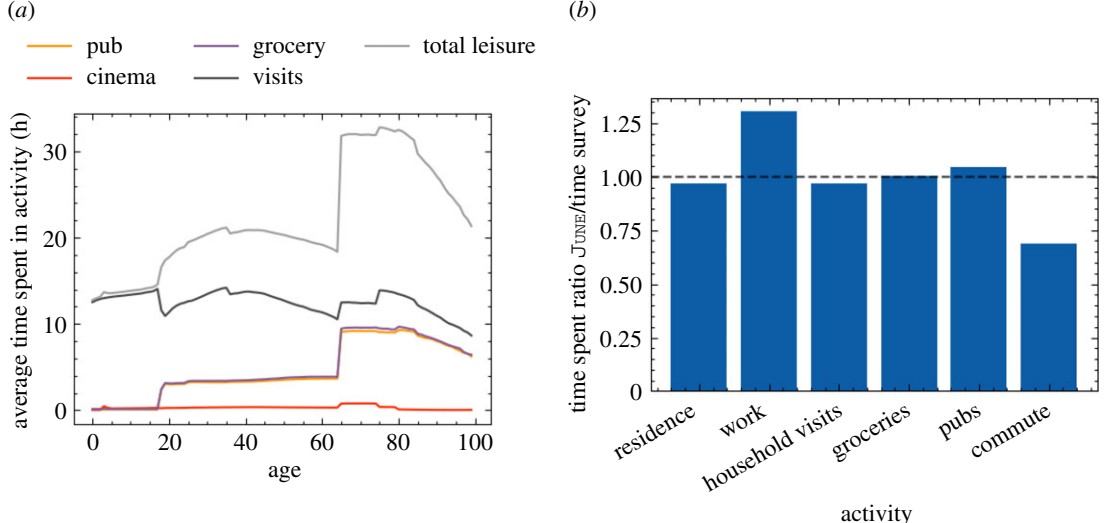

**Figure 6.** Leisure activities in JUNE. (*a*) Time spent in leisure by age in JUNE. (*b*) Comparison of the fraction of time spent in different activities in JUNE and the time survey.

If the individual is selected to participate in one of these activities, the chosen activity, $i$, is then selected according to its probability

$$\mathcal{P}_i = \frac{\lambda_i(\{p\})}{\sum_{j=1}^N \lambda_j(\{p\})}.$$

(4.2)

The person is then moved to the relevant location corresponding to this activity. If no activity is selected, the individual will stay at home.

A summary of how much time is spent each week on various activities as a function of age is reported in figure 6*a*. In figure 6*b*, we show a comparison of the amount of time spent at home, work, grocery shopping, eating at restaurants/pubs and commuting, between JUNE and the UK Time Use Survey, 2014–2015 [31]. Care home and cinema visits are not accounted for in the time survey.

## 4.2. Localized activities

Within JUNE social activities outside the static home, work and school settings can be specified and given their own specific interaction model. Indeed, collections of sub-models can be specified, with substitutable activity choices. For instance, for the English population, these activity models are informed and parametrized by time surveys available from the ONS [29], which identify a variety of activity types including time spent at home, work or in school. In addition, we have identified five additional settings which we assume are similarly relevant for the spread of the disease and have a similar level of social mixing: visits to pubs or restaurants (pubs), cinema visits (cinemas), shopping (groceries), visiting friends or relatives in their homes (household visits), or visiting family members in care homes (care home visits).

For England, we have located 120 000 pubs and restaurants according to their geo-coordinates, as well as 32 000 stores and 650 cinemas, with data from OpenStreetMap [32]. Each time a person is assigned to any of 'pubs', 'groceries' or 'cinemas', we pick a random venue from the $n$ venues closest to their place of residence, or the closest venue if the distance to any of them is greater than 5 km. We have chosen $n = 7$ for pubs, $n = 15$ for shopping stores and $n = 5$ for cinemas. Note that there are no permanent 'workers' in these venues who return to a single venue daily; only 'attendees' who choose their venue at random. Further locations such as gyms and places of worship can be easily added to the activity model, and, of course, it can easily be adjusted to other societies.

In addition, we model interactions in naively constructed social networks, by linking each household to a list of up to $\mathcal{N}$ other households in the same super area. One of the households in this list is selected if 'household visits' is chosen as activity during a time-step. Residents will stay at home to receive the incoming visitor, who in turn may also bring their whole household with them according to a probability described by an external parameter. Comparison with national surveys suggests that setting the number of linked households $\mathcal{N} = 3$ provides realistic movement profiles. While care home residents in JUNE

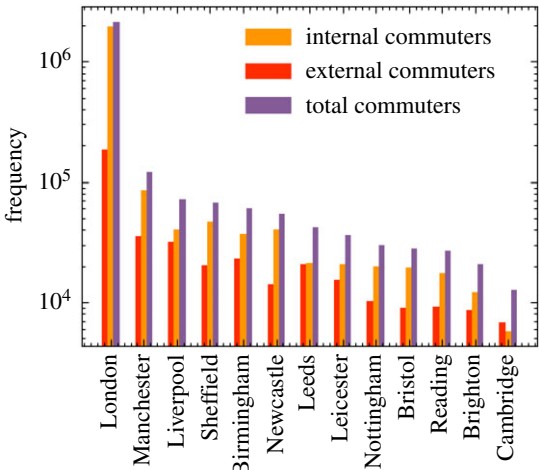

**Figure 7.** Number of internal and external commuters by city as modelled in June.

cannot visit other people, each resident is connected to a household of them in the local super output area from whom they can receive visitors. June also models the interactions that result from elderly people needing help in their daily activities. Each person older than 65 years old has a probability, increasing by age, of needing some kind of assistance in their daily activities. We therefore assign a member of the local super area to be the carer of an elderly person, following the data available in [33]. Every weekday, the carer spends their leisure time visiting the household of the person needing domestic care.

## 4.3. Modelling mobility: commuting patterns

Mobility is modelled in June through a number of transport types that collect and move the virtual population within a pre-specified region or connecting regions within a simulated country. June permits an arbitrary number of transport networks of different types with different interactions (e.g. bus networks, train networks and road networks). For any given movement of the population June ensures that each individual is singularly accounted for with an equivalent end or return location. Travellers move between nodes on these transport networks, and may share their means of transportation and potentially interact in a time consistent manner.

To model commuting and rail travel in England we use data provided by the UK Department for Transport [34]. Large metropolitan areas are selected as the major transit node for the network. Commuting induces social mixing between many people who may not normally come into contact and reflects the importance of transport as a mechanism for promoting the geographical spread of infection supplementing the spread from individuals moving to a new location and infecting other individuals at that location.

To fill our origin–destination matrix we use information contained in the ONS database concerning the mode of commuting of individuals at the area (OA) level [35], to distribute commuting modes probabilistically. We define two modes of public transport, 'external' which defines those commuting in and out of metropolitan areas, and 'internal' which defines those commuting within these areas. Metropolitan areas are defined using data obtained from the ONS [36]. For the sake of computational efficiency, we model only the travel patterns of those working inside metropolitan areas, who in fact represent the overwhelming majority of public transport commuters. This includes commuters who live and work in the city, as well as those who are entering the metropolitan area from outside. The number of internal and external commuters by city in England is given in figure 7. The cities included are geographically spread across England thereby accounting for major commuting patterns in most regions modelled. In total, we explicitly model commuting into 13 out of a possible 109 cities in England, which accounts for 60% of all metropolitan commuters and 46% of all those using public transport to commute to work. Figure 8 shows maps of the residences of internal and external commuters in two cities in our model, where the inner section in white denotes the respective metropolitan areas. Specifically, from figure 8b, we can see that, given the large commute radius of cities like London (we observe a similarly large radius for Birmingham and several other cities), commuting can be a key driver for the inter-regional spread of infectious diseases.

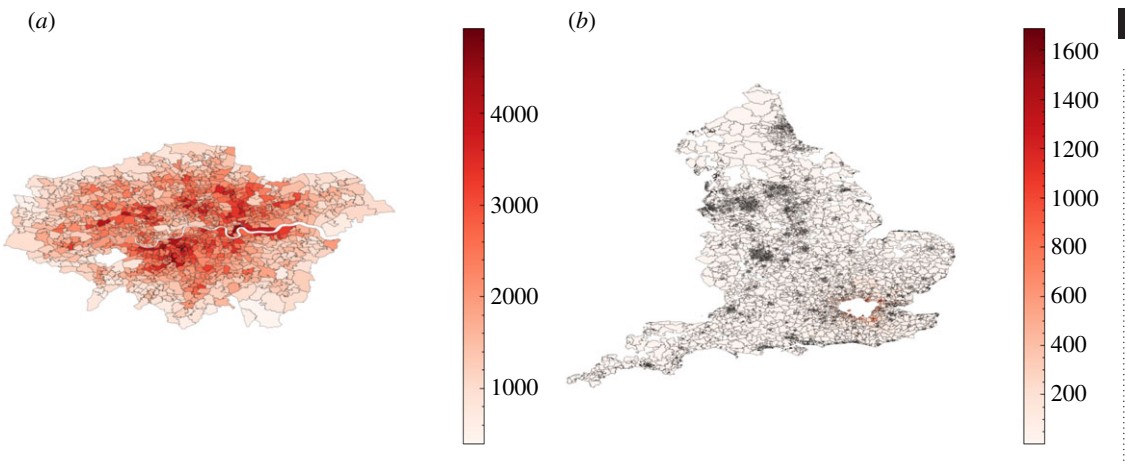

**Figure 8.** Commuting maps for London as derived from June. Any visible super area (MSOA) which is not completely white has at least one commuter from that location. (*a*) Number of internal commuters in London. (*b*) Number of external commuters in London.

Travelling within a metropolitan area, i.e. the internal commuting mode, is modelled as a self-connected loop—practically speaking this means that internal commuters may in principle interact, irrespective of the actual movement inside the city. For external commuting, the travel into and out from the metropolitan area, we identify shared routes for commuters living in neighbouring areas and super areas. The number of possible routes into each city, and therefore the number of ways to divide regions around the cities, is informed by the approximate number of rail network lines into each city—currently this is set to eight in London and four for each of the other 12 cities [37].

We randomly partition people sharing the same commuting route into subgroups, 'carriages', which define the environment in which social interactions take place. The commuting time-step is run twice a day and in each run the travellers are randomly distributed into carriages. The number of people per carriage is determined by city-dependent data obtained from the UK Department for Transport [34]. More details on the specific algorithm for modelling commuting in June can be found in appendix A.4.

## 4.4. Social interaction frequencies and intensities

Social contact matrices [10,15] provide information about the age-dependent frequency and intensity of in-person contact in different social settings, an important ingredient to many epidemiological simulations. They measure the average daily number of conversational and physical contacts between individuals of different ages. This means that they are normalized to the size of the population in the respective age bins, but do not account for whether they can take part in such contacts. To use them within June, we therefore have to account for the fact that social settings define the group of people coming into contact with each other. To exemplify this, consider the case of contacts between adults and students in schools. While the social contact matrices in the literature normalize the number of contacts of a 30-year-old with children of a certain age to the number of 30-year-old adults in the population, in June only a subset of 30-year-old adults work as teachers and can therefore interact with the children. In the construction of matrices specific for June, we therefore combine the results from [10,15] with simple assumptions about possible participants in contacts.

Averaging over age ranges in different settings, we arrive at simplified social mixing matrices, $\chi_{si}^{\mathscr{L}}$, which will be comparable to the inputs from literature upon combination with the model results for the composition of social environments. Below we list our simplified social mixing matrices inferred from the literature, with $\mathscr{L} \in \{(H), (S), (W)\}$ (home, school, workplace), as well as the relative proportions, $\phi_{si}^{\mathscr{L}}$, of physical contacts. The latter are relevant, since in line with standard approaches, closer physical contact in June is proportional to a higher propensity for transmission for the etiological agent.

For the households social mixing matrices, we define four categories, young children (K), young dependent adults of age 18 or more (Y) that still live with their parents, adults (A), and older adults (O) of age 65 and over. We use

$$\chi_{ij}^{(H)} = \begin{pmatrix} 1.2 & 1.69 & 1.69 & 1.69 \\ 1.27 & 1.34 & 1.47 & 1.50 \\ 1.27 & 1.30 & 1.34 & 1.34 \\ 1.27 & 1.50 & 1.34 & 2.00 \end{pmatrix} \quad \text{and} \quad \phi_{ij}^{(H)} = \begin{pmatrix} 0.79 & 0.70 & 0.70 & 0.70 \\ 0.70 & 0.34 & 0.40 & 0.40 \\ 0.70 & 0.40 & 0.62 & 0.40 \\ 0.70 & 0.40 & 0.40 & 0.56 \end{pmatrix}. \quad (4.3)$$

For household visits, we make the simplifying assumption that the same matrices also describe the contacts between visitors and residents. For visits to care homes, we believe that visitors come into contact only with residents and care home workers, and not with other visitors. We therefore hypothesize six conversational contacts with residents and 1.5 with care home workers.

Social contacts in schools identify teachers (T) and students (S); the latter are organized in year groups and further divided into classes of up to 40 students. In our age-averaging, we implicitly assume that the number and character of teacher–student contacts is independent of the age of the students. Student–student contacts are assumed to be most frequent within a class or year group, and fall off steeply with the age difference. This behaviour is captured by fitting a matrix with values for the age-diagonal elements and a fall-off per year age-difference by a factor of 3. Therefore we have

$$\chi_{ij\in\{T,S\}}^{(S)} = \begin{pmatrix} 4.8 & 0.75 \\ 15 & \chi_{SS}^{(S)} \end{pmatrix} \quad \text{and} \quad \phi_{ij\in\{T,S\}}^{(S)} = \begin{pmatrix} 0.05 & 0.08 \\ 0.1 & \phi_{SS}^{(S)} \end{pmatrix}, \tag{4.4}$$

with the student–student matrices taking the following form

$$\chi_{SS}^{(S)} = \begin{pmatrix} 2.5 & 0.75 & 0.25 & \dots \\ 0.75 & 2.5 & 0.75 & \dots \\ 0.25 & 0.75 & 2.5 & \dots \\ \vdots & \vdots & \vdots & \ddots \end{pmatrix} \quad \text{and} \quad \phi_{SS}^{(S)} = 0.15 \; \forall i, j \in \{S\}. \tag{4.5}$$

For the contacts at work, we do not take into account of any age-dependence and, in the absence of data, do not model any sector-dependent variation of their number of intensity, thus

$$\chi_{si}^{(W)} = 4.8 \quad \text{and} \quad \phi_{si}^{(W)} = 0.07. \tag{4.6}$$

In appendix C, we detail the algorithms used to construct the social mixing matrices used in JUNE including the matrices for other locations not listed here.

These social mixing matrices in JUNE are defined for a setting-specific characteristic time $t_{\text{char}}$, so the total number of contacts in a time interval $\Delta t$ in a given setting is then modified by a factor $\Delta t / t_{\text{char}}$.

To validate these simplified matrices, we include them within JUNE where they are combined with the composition of the specific social settings. In figure 9, we show the resulting contact matrices as 'measured' from the JUNE simulation. The effect of the combination with the composition is most pronounced in the household matrices which exhibit textures that can be directly traced back to the age intervals of children, dependent children/young adults, adults and older adults that JUNE inherits from the ONS data. These matrices naturally recover much of the structure present in those recorded in [10,15]. Further details on the methodology for extracting these matrices from JUNE can be found in appendix C.5.

# 5. Infection modelling: spreading and health impact

The transmission of infection through social interactions described in the `interaction` layer, and the progression of the disease and its impact on the individual, are both modelled in the `disease` layer. Although we focus on the case of COVID-19 here, this layer is designed to be generalizable and can contain more than one circulating etiological agent and or types of agent.

Throughout this section and the rest of the paper, we will use two definitions of COVID-19 'cases'. The first is when we refer to cases in the model itself—here, a case of COVID-19 is an infected agent which may be symptomatic or asymptomatic. The second is when referring to cases in reality—here, a case is someone who has tested positive for COVID-19. Since the latter is subject to testing coverage, capacity and efficacy, we do not use these for fitting or validation purposes.

## 5.1. Infection transmission

JUNE models the transmission of an infection from infecting individual, $i$, to susceptible individual, $s$, in a probabilistic way. The probability of infection in a social setting within a group of people, $g$, at a location, $L$, depends on a number of factors:

— the number, $N_i$, of infectious people $i \in g$ present;
— the infectiousness of the infectors, $i$, at time $t$, $I_i(t)$;
— the susceptibility, $\psi_s$, of the potential infectee, $s$;

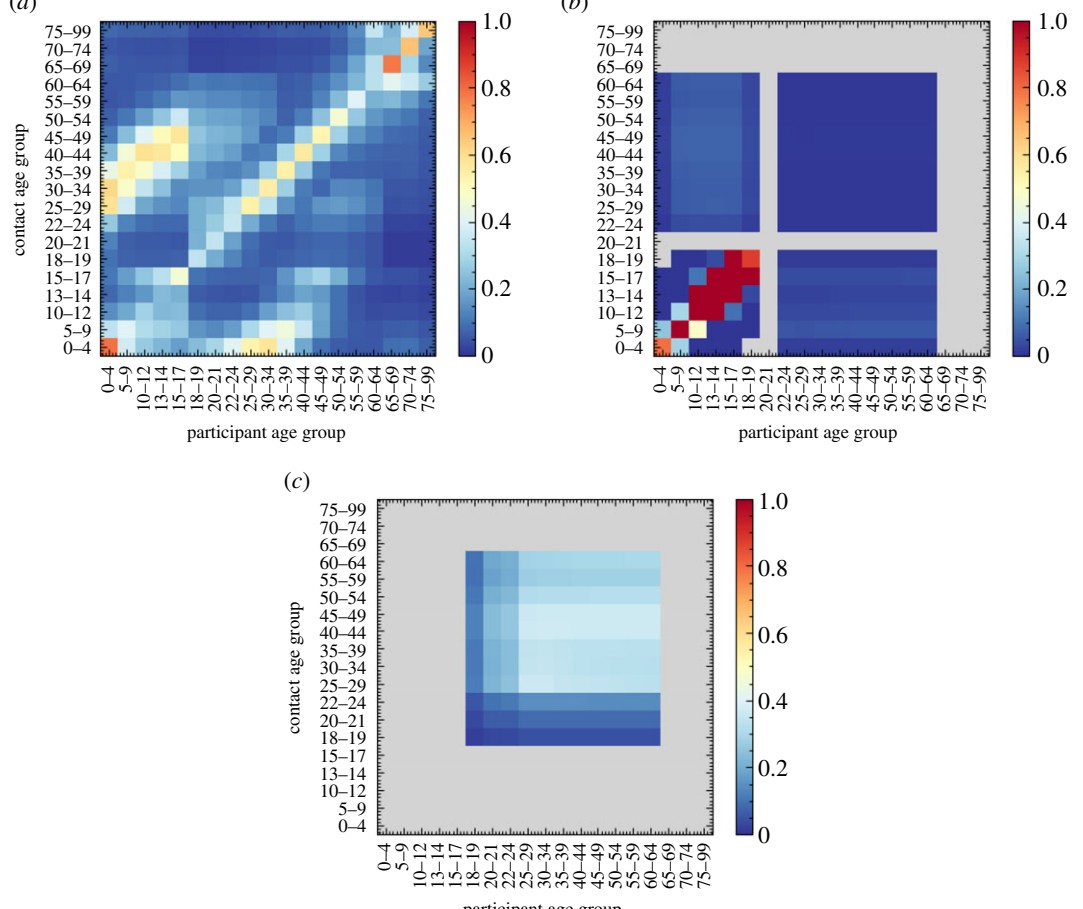

**Figure 9.** Social contact matrices for England derived from June, before any mitigation strategies are implemented. Colour bars show (average) number of contacts in social settings between age groups, with all colour scales truncated at one to show differences between settings, while still clearly showing the structure in the matrices. (a) Household, (b) school, (c) company.

— the exposure time interval, $[t, \ t + \Delta t]$, during which the group, $g$, is at the same location;
— the number of possible contacts, $\chi_{si}^{(L)}$, and the proportion of physical contacts, $\phi_{si}^{(L)}$, in location $L$, both taken from equations (4.3)–(4.6) in §4.4;
— and the overall intensity, $\beta^{(L,g)}$, of group contacts in location $L$.

Most of these ingredients depend on the time, $t$, of the contact. For example, the number of contacts, $\chi_{si}^{(L)}$, and the proportion of physical contacts, $\phi_{si}^{(L)}$, and the overall contact intensity, $\beta^{(L,g)}$, will change with the implementation of social distancing policies. To simplify notation, we introduce a combined contact intensity for a group $g$ with size $N_g$ at location $L$,

$$\beta_{si}^{(L,g)}(t) = \beta^{(L,g)} \cdot \frac{\chi_{si}^{(L)}(t)}{N_g} \left\{ 1 + \phi_{si}^{(L)}(t)[\alpha(t) - 1] \right\}, \tag{5.1}$$

where the ratio $\chi/N_g$ provides a simple parametrization of the probability of $s$ being in contact with another individual in the group, and $\alpha(t) > 0$ describes the relative impact of close physical contacts. Both the factor $\alpha(t)$, which we assume to be the same for all locations, and the location- and group-specific contact intensities, $\beta^{(L,g)}$, are taken from fits to data.[7]

In the construction of an infection probability for a susceptible individual, $s$, we make a number of assumptions. First of all, we model the probability of being infected as a Poisson process. In keeping with the probabilistic process, the argument of the Poisonnian is given by a sum over individual pairs

---

[7]In reality, the $\beta$ parameters are fitted in a location-specific way, irrespective of the group—i.e. a location of type $L$ in containing one group of people, $g_1$, and another location of the same type, but with a different group of people, $g_2$, (e.g. two pubs in different places) will have the same $\beta$.

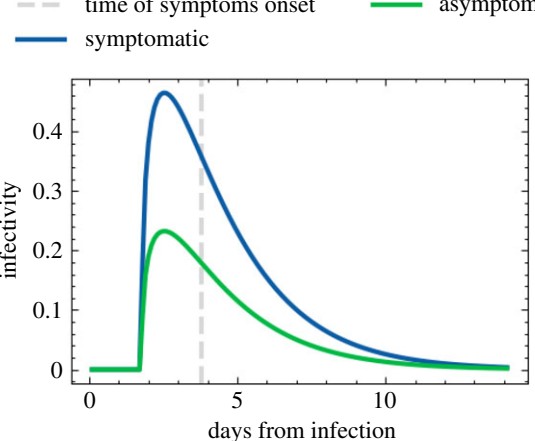

**Figure 10.** Time-dependent infectiousness profile, $f_I(t')$, shown for the same realization of the infection but where the infected person is symptomatic or asymptomatic.

of infectious individuals with the susceptible person, implying a simple superposition of individual infectiousness. The underlying individual transmission probabilities are written as the product of the susceptibility of the susceptible individual, the infectiousness of the infected person, and the contact intensity, all integrated over the time interval in which the interaction occurs. The integration over time ensures that the transmission probability increases with the time of exposure. We therefore arrive at the transmission probability, i.e. a probability for $s$ to be infected as

$$\bar{\mathcal{P}}_s(t,\ t + \Delta t) = 1 - \exp\left[-\psi_s \sum_{i \in g} \int_t^{t+\Delta t} \beta_{si}^{(L,g)}(t')\mathcal{I}_i(t')\,\mathrm{d}t'\right]. \tag{5.2}$$

Note that in the actual implementation, we approximate the integral over time with a simple product,

$$\int_t^{t+\Delta t} \beta_{si}^{(L,g)}(t')\mathcal{I}_i(t')\,\mathrm{d}t' \longrightarrow \beta_{si}^{(L,g)}(t)\mathcal{I}_i(t)\Delta t. \tag{5.3}$$

This leaves us to fix the last two ingredients in equation (5.2), the individual susceptibility, $\psi_s$, and the infectiousness, $\mathcal{I}_i(t)$. Contemporary peer-reviewed academic research on susceptibility to infection by the etiological agent with or without the onset of disease symptoms is sparse and inconsistent. Following some evidence, for example in [38] and [39], on transmission and susceptibility of children (using the UN classification), we fix $\psi_s = 0.5$ for children under the age of 12, and $\psi_s = 1$ for everybody else. The infectiousness of individuals, $\mathcal{I}_i$, changes with time, and it is not directly measurable. To model its behaviour, we use the temporal dependence of viral shedding as a proxy for infectiousness. Studies in the context of COVID-19 have shown that viral shedding peaks at or slightly before the onset of symptoms, and then begins to decrease [40]. In JUNE, we use a globally defined temporal dependence of infectiousness, $f_I(t)$, and multiply it with a peak value, $\mathcal{I}_{i,\max}$, which depends on the infected individual,

$$\mathcal{I}_i(t') = \mathcal{I}_{i,\max} \cdot f_I(t'). \tag{5.4}$$

We choose the maximal infectiousness according to a log-normal distribution parametrized by its median $\exp(\mu) = 1$ and shape $\sigma = 0.25$. The long right tail of the log-normal distribution allows for small numbers of highly infectious individuals more likely to precipitate superspreading events (SSEV). We also capture the conjectured reduced infectiousness of individuals with no or only mild symptoms. Following a similar parametrization to that in [41], we multiply the maximal infectiousness of asymptomatic individuals by 0.5. In figure 10, we show an example of the time evolving profile for an infected individual in JUNE, comparing the resulting infectiousness for different symptoms. For the time-dependent profile, we use the gamma distribution as fitted in [40],

$$f_I(\tau = t' - t_0 - t_{\mathrm{inc}}, a) = \frac{\tau^{a-1}\,\mathrm{e}^{-\tau}}{\Gamma(a)}, \tag{5.5}$$

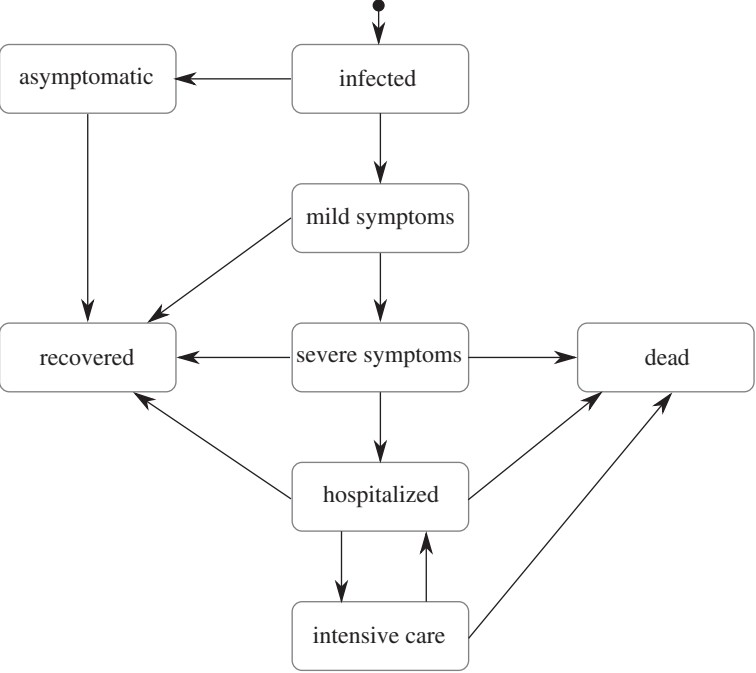

**Figure 11.** Pathways for the infection progression and possible outcomes. Note that in our model a patient can only go to the intensive care once, and that a patient that returns from the intensive care to the hospital will survive.

where $t_0$ is the time of infection, $t_{inc}$ is the incubation period, sampled from a normal distribution centred at 2 days prior a possible onset of symptoms and with a width of half a day, $a$ is the shape parameter of the gamma distribution, and $\Gamma(a)$ is the the gamma function.

## 5.2. Infection progression

When an individual is infected, they will experience different impacts on their health. Figure 11 presents the paths available in June for the progression of the infection that aim to capture different symptom severities, outcomes, and their operational impact on the healthcare system, i.e. whether patients are hospitalized or admitted to intensive care or treatment units (ICU/ITU). Once an individual is infected, June selects their specific complete path according to these probabilities. These paths are codified as a sequence of possible different stages of the disease (infected, asymptomatic, mild, severe, hospitalized, ICU/ITU, dead, recovered) in addition to characteristic time intervals for each stage. The latter are chosen randomly according to probability functions informed by available data. The paths terminate with the individuals either dead and taken out of the simulation, or recovered, in which case their susceptibility is set to 0, making them immune to reinfection.[8]

June distinguishes the following different routes for the progression of the infection with rates depending on the characteristics of the infected individual (currently age, sex), summarily denoted by $p$:

1. asymptomatic individuals, rate $R_{I \to A}(p)$, continue their life normally;
2. individuals with mild symptoms, rate $R_{I \to M}(p)$, usually continue their lives as normal, except if certain policies are activated;
3. individuals with severe but not lethal symptoms, rate $R_{I \to S}(p)$, stay at home until recovery;
4. individuals with severe symptoms who will eventually die in their residences, with rate $R_{I \to DR}(p)$;
5. individuals who are admitted to hospital but will recover, with rate $R_{I \to H}(p)$;
6. individuals who are ultimately admitted to ICU/ITU before recovering, with rate $R_{I \to ICU}(p)$;
7. individuals who are admitted to hospital and will die there, with rate $R_{I \to DH}(p)$ and
8. individuals who are admitted to ICU/ITU and die there, with rate $R_{I \to DICU}(p)$.

The determination of probabilities for the different paths is based on COVID–19 data that are not entirely sufficient to develop a complete and detailed picture. As a consequence, we supplement them with

---

[8]June could also model reinfection of individuals but to date there are no data constraining this in the context of COVID–19.

**Table 2.** Datasets used in the derivation of mortality and hospitalization rates. GP stands for people living in a household, and CH stands for people living in care homes. If not specified, datasets involve people from both populations. All data is taken until 13 July 2020, consistently with the seroprevalence study [42].

| quantity | source |
| --- | --- |
| population by age, sex and residence type | [47,48] |
| seroprevalence in GP by age | [42] |
| seroprevalence in CH by age | [44] |
| deaths by place of occurrence and residence type | [43] |
| deaths profile by age and sex | [43] |
| deaths in CH profile by age and sex | [49] |
| hospital deaths profile by age, sex | [50] |
| hospital deaths in CH profile by age, sex | [51] |
| ICU/ITU deaths profile by age, sex | [51] |
| total hospital admissions | [52] |
| hospital admissions profile by age, sex | [50] |
| ICU/ITU admissions profile by age, sex | [51] |
| hospital admissions in CH profile by age, sex | [51] |

assumptions by inferring some properties through cross-relating datasets. In the following, we will outline our procedure which is largely predicated by our choice of the example at hand—the spread of COVID-19 in England. We will use a notation where $N_X(p)$ denotes the number of cases satisfying criterion $X$ for people with characteristic properties $p$.

The construction of reasonable progression paths, and their probabilistic distribution, relies critically on the knowledge of how many people have been infected, as well as the dependence on attributes such as age and sex. COVID-19 tests between February and May 2020 in the UK were mostly administered to people presenting symptoms or people that have been in close contact with confirmed cases in hospital, thereby biasing the results. We therefore need to infer the number of infections from other controlled studies, such as antibody tests. In [42], the seroprevalence, $r_{sp}(p)$, of COVID-19 in the adult population in England was determined through a sample of more than 100 000 adults, showing a reduction in seroprevalence with increasing age. Because the seroprevalence is an estimate of all people that were infected up to the time of the test and—most importantly—survived, we need to correct for those who died of the disease until this point. This turns out to be an important correction, especially in older age bins due to the non-negligible probability of elderly who died. We therefore add the age- and sex-dependent number of deaths, $N_D(p)$, reported by the ONS [43], to the corresponding numbers inferred from the seroprevalence to arrive at the total number of cases, $N_{tot}(p)$,

$$N_{tot}(p) = r_{sp}(p)N(p) + [1 - r_{sp}(p)]N_D(p), \tag{5.6}$$

where $N(p)$ is the total population number in England with characteristics $p$. We note that there were two population groups excluded from the serology survey: people under the age of 18, and care home residents. For the former, we assume that their seroprevalence by age is identical to the population group aged 18, while for the latter, we set a flat seroprevalence by age at 11% value as reported in the Vivaldi report of the UK Department of Health and Social Care [44] in the beginning of July 2020.

Health outcomes given a simulated infection are captured in $R_{I \to X}$, where $X$ is one of the eight trajectories listed in figure 11. The asymptomatic rate, $R_{I \to A}$, and the mild case rate, $R_{I \to M}$, are taken from a calibration done in [41] from [45,46]. To calculate the different hospitalization and fatality rates, we have used a series of datasets listed in table 2, all of them containing data until 13 July 2020, to be consistent with the considered seroprevalence values. In order to avoid possible irregularities in our results derived from the use of different data sources, we normalize all our death data to the ONS reported numbers of total deaths (51 443), hospital deaths (32 164) and residence deaths (19 279), [43] and then use more granular data to distribute deaths by age and sex for each place of death occurrence [50,51]. Likewise, the total number of hospital admissions is taken from [52], and distributed by age, sex and residence type also using [50,51]. The number of deaths in care homes reported in [53] is only

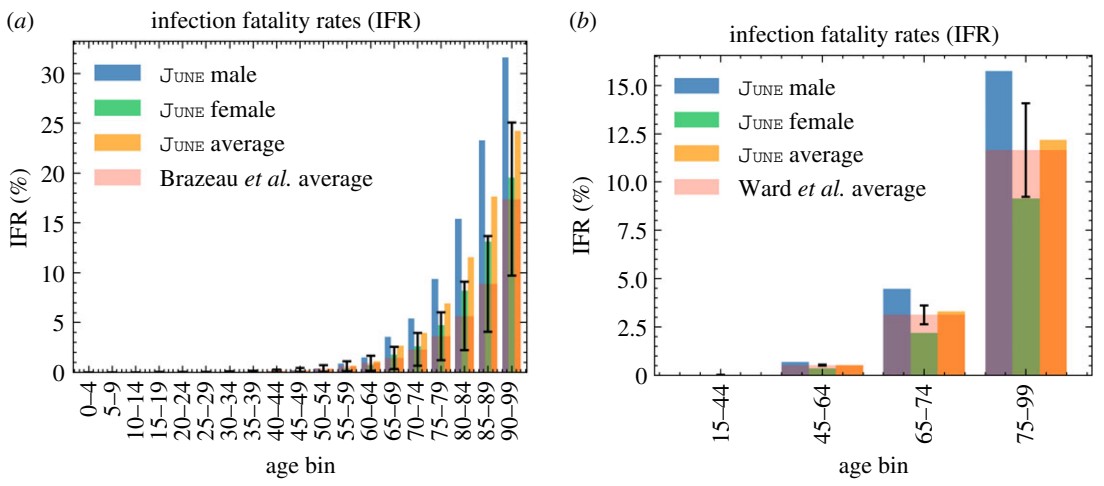

**Figure 12.** IFR comparison of June with various estimates of community transmission. Error bars show 95% CI on the IFRs as estimated from data. (*a*) IFR comparison of June with [54], (*b*) IFR comparison of June with [42].

reported by age until late June, so we assume that the distribution does not change until 13 July 2020. We also ensure that we correctly account for differences in reporting times. As a first step, we calculate the overall infection fatality rate (IFR) for the general population outside care homes (GP),

$$R_{I \to D}^{GP}(p) = \frac{N_D(p) - N_D^{ch}(p)}{N_{tot}(p) - N_{tot}^{ch}(p)}, \tag{5.7}$$

which can be directly compared with the results from the REACT2 study [42] (figure 12*b*), and the Imperical College London COVID-19 report 34 [54] (figure 12*a*). The remaining rates just follow from the same methodology,

$$R_{I \to X}^{GP}(p) = \frac{N_X(p) - N_X^{ch}(p)}{N_{tot}(p) - N_{tot}^{ch}(p)} \tag{5.8}$$

and

$$R_{I \to X}^{CH}(p) = \frac{N_X^{ch}(p)}{N_{tot}^{ch}(p)}, \tag{5.9}$$

where X refers to one of deaths or hospital admission in the normal hospital ward or in the ICU/ITU. The rate of non-hospital deaths is computed by subtracting the hospital death rates from the overall IFRs. Finally, the probability of having severe symptoms but recovering at home is given by

$$R_{I \to S}(p) = 1 - \sum_{i \neq S} R_{I \to X_i}(p). \tag{5.10}$$

The results of computing the individual infection outcome rates by age, sex and residence type are shown in figure 13. The most important visible difference is the disparity on the fatality rates between care home residents and the general population. This could be the reflection of various reasons, including, for example, a generally poorer health condition of the care home population, or differences in admission policies to hospitals. Consistent with the ONS data [53], most of the care home deaths occur within the care home residence itself, while the probability of being admitted to the hospital decreases with age. Likewise, both for the general population and the care home population, people aged 55–70 years old are the group most likely to be admitted in the ICU/ITU. Females are less likely in general to develop a severe infection of COVID-19, with fatality rates roughly equivalent to those of a male 5 years younger.

Once an infection outcome has been determined, the infected individual follows a symptoms trajectory composed of different stages. The time spent at each stage is sampled from different distributions derived from different data sources. In table 3, we list the different stages per trajectory by infection outcome, and the details on the various timings are listed in appendix D. In figure 14*a*, we show the probability density functions for the incubation time, and the time to die or recover in hospital.

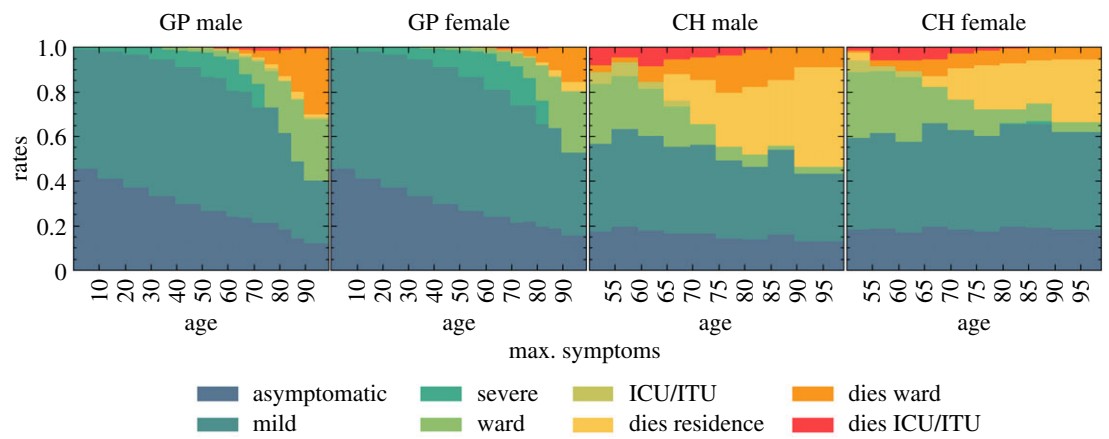

**Figure 13.** Rates of different infection outcomes for males and females living in households and care homes. For care home residents, we only show the rates for people aged over 50, as the younger ones are assumed to follow the general population rates.

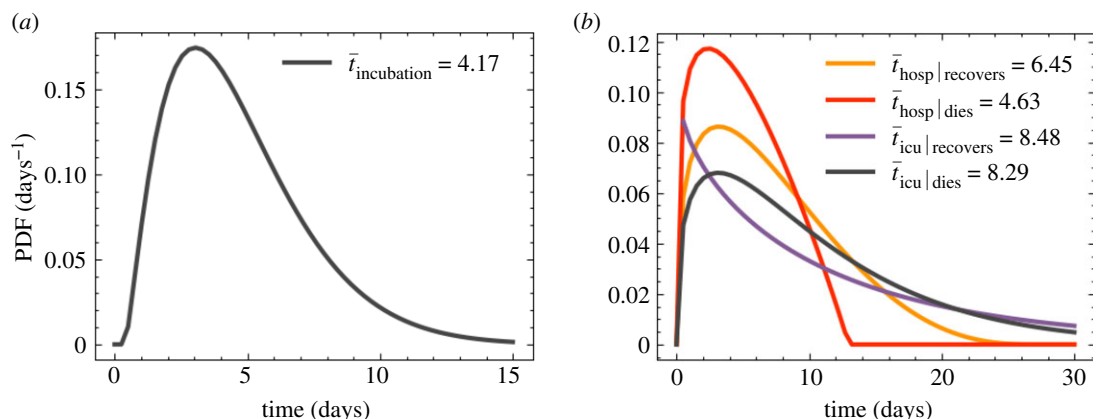

**Figure 14.** Probability density functions for symptom and progression timing. (*a*) Time taken for an infected individual to develop symptom. (*b*) Time spent in hospital by patients given their infection.

**Table 3.** List of different trajectories through disease progression, with stages and, in brackets, the distribution from which corresponding timings are drawn. For their definition see table 7. The available stages are **I**nfected, **A**symptomatic, **M**ild and **S**evere symptoms, admitted to a regular **H**ospital or an **ICU/ITU** ward, and, finally, as outcomes, **R**ecovered or **D**ead.

| trajectory | stages | | | | | | |
|---|---|---|---|---|---|---|---|
| asymptomatic | $I[\beta_I]$ | $A[C_{14}]$ | | | | | R |
| mild | $I[\beta_I]$ | $M[C_{20}]$ | | | | | R |
| severe | $I[\beta_I]$ | $M[C_{20}]$ | $S[C_{20}]$ | | | | R |
| death at home | $I[\beta_I]$ | $M[LN_M]$ | $S[C_3]$ | | | | D |
| ward | $I[\beta_I]$ | $M[LN_M]$ | $H[\beta_H]$ | $M[C_8]$ | | | R |
| death in ward | $I[\beta_I]$ | $M[LN_M]$ | $H[\beta_D]$ | | | | D |
| ICU/ITU | $I[\beta_I]$ | $M[LN_M]$ | $H[LN_{ICU}]$ | $ICU[e_{ICU}]$ | $H[e_H]$ | $M[C_3]$ | R |
| death in ICU/ITU | $I[\beta_I]$ | $M[LN_M]$ | $H[LN_{ICU}]$ | $ICU[e_D]$ | | | D |

## 5.3. Seeding infections

In the absence of sufficiently detailed knowledge of how epidemics arrive in a country, we seed infections using secondary information such as the number and regional distribution of observed cases. In the example of the simulating the spread of COVID-19 in England, we use the number of

COVID-19-related deaths recorded in hospitals to estimate initial infection numbers and their regional distribution. Accounting for the time delay between infection and possible death, and for the probability of admitted patients to die, we have

$$N_{\text{tot}}(t, x) = \frac{1}{\bar{R}_{\text{H}\to\text{D}}(x)} \quad N_{\text{H}\to\text{D}}(t + \Delta t_{\text{D}}, x), \tag{5.11}$$

where $N_{\text{tot}}(t, x)$ is the estimated number of cases in a region, $x$, on day, $t$, $N_{\text{H}\to\text{D}}(t, x)$ is the number of observed deaths in the region at date $t$, and $\bar{R}_{\text{H}\to\text{D}}(x)$ is the rate for people dying in hospital in the region $x$, where the average over the characteristics $p$ is given by

$$\bar{R}_{\text{H}\to\text{D}}(x) = \frac{1}{N_{\text{H}\to\text{D}}(t + \Delta t_{\text{D}}, x)} \sum_{i \in N_{\text{H}\to\text{D}}(t+\Delta t_{\text{D}},x)} R_{\text{H}\to\text{D}}(p_i). \tag{5.12}$$

The relatively large statistical fluctuations in the initial phase of an epidemic, and possibly differing time profiles across regions, translate into the need for a region-specific seeding. This difference is highlighted by contrasting the seeding for London, where we introduce initial infections over two days only (28–29 February 2020) with the northeast of England and Yorkshire, where we seeded infections for a week, 28 February–5 March 2020. We introduce the estimated number of daily cases in each of the regions until the following criterion is met,

$$N_{\text{tot}}(t < T(x), x) > 0.1 N_{\text{tot}}(t_{\text{max}}, x), \tag{5.13}$$

where $T(x)$ is the number of days over which we seed new infections in region $x$, and $N_{\text{tot}}(t_{\text{max}}, x)$ is the maximum number of cases that region $x$ would reach in any given day, estimated from the maximum number of daily deaths in hospital. It is important to define the seeding for the infection based on the maximum number of cases each region will have, since the different regions are experiencing different stages of the epidemic at any given time.

# 6. Mitigation policies and strategies

Policies and interventions, often enacted by governing bodies, are introduced in an attempt to mitigate and control the spread of infectious diseases. In general, such policies are highly dependent on the type of infection and social norms in the affected population, and may include guidelines on how to change individual patterns of behaviour or the closure of certain venues where transmission is estimated to be highly likely. The modular nature of June allows policies to be dynamically activated and deactivated at different points in time to allow for changes in policy decisions. Due to June's granularity, these policies can be implemented at a highly localized level: by type and place of social interactions, by geographical region, by industry sector or venue type. June can also model the population's compliance with the measures, again with high granularity. In this section, we present a variety of policies which can be implemented in June and exemplify their application through those measures that have been enacted by the UK Government to mitigate the spread of SARS-CoV-2.

## 6.1. Behavioural changes

There are a variety of changes in behavioural patterns that are designed to reduce the probability of viral transmission, ranging from simple social distancing, increased hygiene and mask wearing, to quarantining of infected individuals or those who have been in sufficiently close contact with them, and the shielding of vulnerable parts of the population. We model the impact of the former set of measures, social distancing, increased hygiene and mask wearing, through multiplicative reductions in the location-specific contact-intensity parameters, $\beta^{(L,g)}$, see figure 15 for an example. The impact of compliance with social distancing and other, similar measures can be recorded both nationally and sometimes even in specific locations. This allows us to calculate the reduction in the corresponding intensity parameters as follows:

$$\beta^{(L,g)} = M^{(L,g)} \beta^{(L,g)} \tag{6.1}$$

$$= \left[1 - C^{(N)} \cdot C^{(L)} \cdot (1 - E)\right] \beta^{(L,g)}, \tag{6.2}$$

where $M^{(L,g)}$ is the location- and group-specific modification factor, $C^{(N)}$ is the national compliance (i.e. percentage of the population following guidelines), $C^{(L)}$ is the compliance in a given location or social

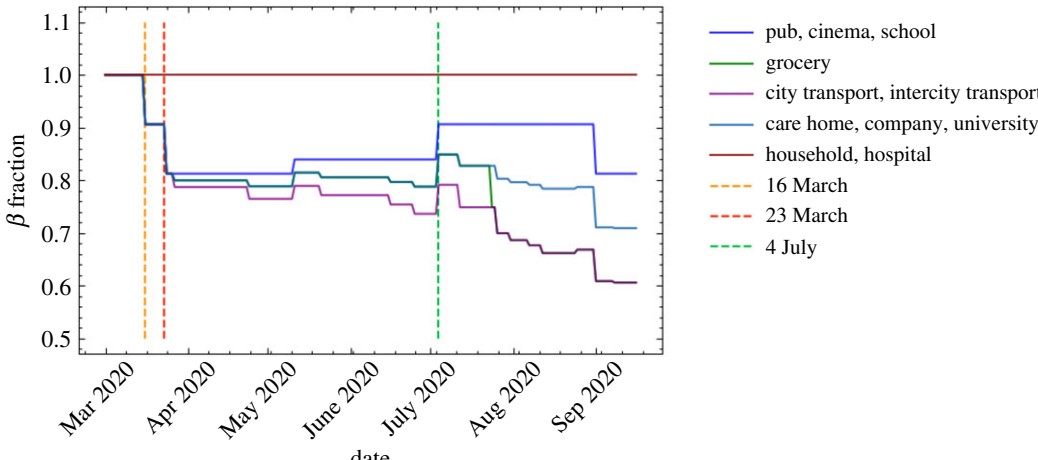

**Figure 15.** Example scenario of different intensity parameters, $\beta^{(L,g)}$, over time normalized to unity (see equation (5.1)). The parameters change due to the effects of compliance with social distancing and mask wearing advice and regulations.

setting $L$, and $E$ denotes the efficiency of the measure. Quarantining is simulated by keeping the individuals in question in their homes for a certain amount of time, and allowing them to interact with members of their household in an otherwise unchanged household setting only. In JUNE, we have the ability to apply different policies to those with mild and severe symptoms, and to quarantine household members of symptomatic individuals. Similarly, JUNE also allows the definition of vulnerable individuals—typically by characteristics such as age—and of a prescription of how shielding policies are enacted relative to this group.

We will now turn to discuss our choices for specific measures. There have been a variety of studies on the effectiveness of social distancing with respect to COVID-19 and other infectious diseases. A comprehensive systematic review and meta-analysis [55] suggested that the relative risk of infection decreases by approximately a factor of 2 per metre distance. In practice, however, the efficiency of social distancing is highly dependent on external factors, in terms of both physical and social environment. We therefore use this literature as a benchmark, assuming on average 1 m social distancing, $E = 0.5$, and fit the effects of social distancing to data where possible (see §6.3).

We simulate mask wearing according to equation (6.2), i.e. by multiplicatively reducing the $\beta$ parameters in different locations. There is a significant body of literature on the effectiveness of mask wearing, including differences based on the material of the mask and the locations in which they are worn [55–57], as well as changes in efficiency due to re-using or washing them [58,59]. In general, we focus on the wearing of masks by non-healthcare workers in settings outside the home and estimate mask effectiveness, $E$, to be 50% [60], irrespective of the specific location. However, after adjustments for compliance the actual, intensity parameter reduction may be much lower than this, which leads us to believe that this represents a conservative estimate.

In JUNE, quarantining of infected people with mild or severe symptoms is relatively straightforward: afflicted individuals do not leave their household for a predefined period of time—usually 7 to 14 days—but do not change interaction patterns with the residents in their household. In some versions of quarantine policies, household members must also stay at home and isolate themselves. This is modelled in JUNE along the same lines, only the possibly time-dependent compliance of the population with quarantine measures. Clearly, infected individuals with severe symptoms will always stay at home, until they are either recovered, moved to hospital, or died. It should be noted that quarantine sits on top of other, less individual-driven policy interventions included in JUNE which may restrict movement, such as the closure of companies and leisure venues.

Given the additional danger infectious diseases may pose to the more vulnerable and elderly populations, various policies, usually referred to as 'shielding' can be introduced with an aim to protect these individuals. In JUNE, shielding is realized similar to quarantine: vulnerable individuals—usually defined by their age or other characteristics—stay at home and do not interact with others outside their household. Apart from the definition of relevant characteristics, this only leaves a compliance probability to be introduced, which reduces the participation in any other social settings other than households.

## 6.2. Closure of venues

Mitigation strategies that aim at reducing infection transmission through changes in individual behaviour may have to be further supplemented through partial or complete closure of certain parts of public life such as companies, transport, schools and universities.

Starting with the closure of companies, June can realise this important measure in a sector–specific way. June allows the definition of 'key' and 'furloughed' workers, again in a sector–specific way. While the former represent those parts of the workforce that continue with their essential work as usual, the latter never goes to work and instead is given the chance to take part in available leisure activities or stay at home during regular working hours. For the rest of the workforce, June allows the definition of flexible work patterns by assigning daily probabilities for workers to go to their companies.

School and university closure is handled similarly to the closure of companies in June. However, in the case of schools, we are able to close individual year groups as well as entire schools, and we can identify the children of key workers and have them continue going to school. Since the return to in-person schooling may also be voluntary at certain points, or children may only go to school on certain days of the week, we also can apply a compliance factor at the year group level, which is used to probabilistically determine which children attend school on any particular day.

In addition to the partial or complete closure of companies in some industry sectors and of schools or universities, government policies may also close or limit the number or people attending leisure venues, such as restaurants and pubs, cinemas, or similar. In June, we are able to fully or partially close different types of leisure venues either nationally or at a more local level, down to super areas. Partial closure is enacted through a change in the probabilities that people attend different venues, which is both sex and age disaggregated. Modifications to other leisure activities, such as household visits, are also simple to realize in June, by directly modifying the daily probabilities for such activities to take place.

## 6.3. Policies in the UK

The population, interaction and disease layers of June will have time inhomogeneous states and parametrizations dependent on the public policy response and the response of individuals in changing behaviour as an infection spreads. For ex-post analysis, policies can be imposed on the simulation using a set of policy levers with varying effectiveness. For ex-ante prediction, scenarios of responses to different policy combinations can be real-time of pseudo out-of-sample forecasted.

To simulate, ex-post, the spread of COVID–19 in England we impose a set of policies restricting movement and attempting to reduce transmission. Table 4 lists the operational policy interventions enacted by the UK Government from the beginning of March 2020 to October 2020 in an effort to reduce the spread of SARS-CoV-2.

In order to estimate the effects of social distancing on the epidemiological development of COVID–19, we implement multiple staggered social distancing steps during the first wave of the pandemic between 16 March and 4 July 2020 and then again going into September 2020 as schools and universities begin to fully reopen. We fit the national compliance, $C^{(N)}$, with social distancing between 24 March and 11 May 2020 in the range 20–100% when fitting the rest the parameters (see §7). This is taken to be the harshest social distancing step against and others are determined relative to this fit. The location-specific compliance, $C^{(L)}$, is set to be 100% in all locations during fitting to avoid parameter degeneracy and then altered manually thereafter. No social distancing is assumed between household members. We derived the compliance with mask wearing from a YouGov survey [61], and we further stratify the results by social environment or locations. Specifically, we assume complete (100%) compliance with mask wearing during commuting, 50% in care homes and no compliance in pubs, schools or in the household. Compliance with mask wearing in grocery stores is assumed to be at 50% before 24 July 2020, after which we assume complete compliance given the change in government regulations. Since we already assume low intensity parameters in hospitals due to the significant amount of personal protective equipment (PPE) being worn in these scenarios, we do not apply any additional mask wearing in these settings.

On 16 March 2020, the UK Government encouraged people with COVID-19 symptoms to quarantine in their household for 7 days and all those in their household to quarantine for 14 days from symptom onset. We assume that compliance with this measure varies with time as people become more aware of the dangers of COVID-19. Between the 16 March and 23 March 2020 (i.e. the week leading up to the nationwide 'lockdown') we fit compliance with the quarantine policy of those symptomatic to be between 5 and 45%, and the probability that the rest of the household of a symptomatic individual complies is set to the same fitted value. After 'lockdown' comes into effect, the government tightened these rules to only

**Table 4.** List of policies introduced in England by the UK Government at different points in time.

| date (dd/mm/yy) | policy | implemented |
|---|---|---|
| 04/03/2020 | encourage increased hand-washing | |
| 12/03/2020 | case isolation at home | * |
| 16/03/2020 | voluntary household quarantine | * |
| 16/03/2020 | stop all non-essential travel | ** |
| 16/03/2020 | stop all non-essential contact | ** |
| 16/03/2020 | voluntary working from home | * |
| 16/03/2020 | voluntary avoidance of leisure venues | * |
| 16/03/2020 | encourage social distancing of entire population | * |
| 16/03/2020 | shielding of over-70s | * |
| 20/03/2020 | closure of schools and universities | * |
| 21/03/2020 | closure of leisure venues | * |
| 21/03/2020 | stopping of mass gatherings | ** |
| 23/03/2020 | 'stay at home' messaging | ** |
| 11/05/2020 | multiple trips outside are allowed in England only | |
| 13/05/2020 | encouraged to go back to work if they can while distancing | * |
| 01/06/2020 | meeting in groups of up to 6 outside allowed | ** |
| 01/06/2020 | shielding of over-70s relaxed | * |
| 01/06/2020 | school reopening for Early Year and Year 6 students | * |
| 13/06/2020 | 'support bubbles' allowed | |
| 15/06/2020 | school reopening for Year 10 and 12 students for face-to-face support | * |
| 04/07/2020 | leisure venues allowed to reopen | * |
| 04/07/2020 | household-to-household visits permitted along with overnight stays | * |
| 24/07/2020 | mask wearing compulsory in grocery stores | * |
| 01/08/2020 | shielding is paused | * |
| 01/08/2020 | 'Eat Out to Help Out' scheme introduced | * |
| 31/08/2020 | 'Eat Out to Help Out' scheme ends | * |
| 01/09/2020 | schools and universities allowed to reopen | * |
| 01/09/2020 | 'Rule of 6' introduced | |
| 14/10/2020 | tiered local lockdown system introduced | * |

*Indicates policies directly implemented in the model.
**Indicates policies which are indirectly implemented—i.e. other policies effectively implement this one by default.

leave the house for essential trips and one form of exercise per day. To account for this, we increase the symptomatic and household compliance with quarantine to be double their fitted value. In addition, the UK Government strongly suggested that people over the age of 70 were to shield, from 16 March 2020. As in the case of quarantine, we assume people become more compliant with this policy over time and that the initial compliance with the shielding policy for this age bracket increased from 20% in the first week to 70% afterwards. Indeed, one of the reasons the compliance was set to only 70% even after lockdown is due to the fact that people in this age bracket already have a reduced mobility and interaction potential. A 70% compliance therefore still allows them a small chance to interact with others, e.g. in grocery stores, and any higher compliance figures would mean a complete and unrealistic decoupling of this critical population from any social interactions. The shielding policy initially runs until 1 August 2020 and after which the UK Government paused the policy.

To model the partial or complete closure of industry sectors, it is important to understand the descriptions of key workers provided by the UK Government [62], and match these up with the relevant five-digit SIC codes [26]. This ultimately allows us to deduce the proportion of key workers in each sector and assign the

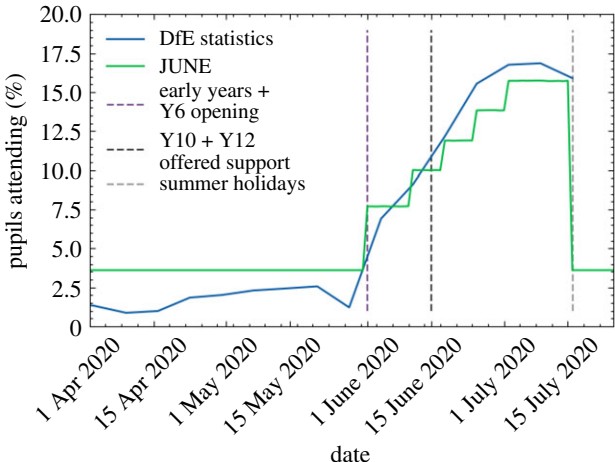

**Figure 16.** School attendance in June compared with data collected by the UK's Department for Education [67].

corresponding key worker attribute probabilistically according to these proportions. In our simulation, we encode findings from the ONS [62], reporting that 33% of the total workforce were key workers in 2019 with 14% able to work from home. We therefore set the proportion of key workers, i.e. those who go to work each day, at 19% of the workforce. We use the same logic to also decide which workers are furloughed in June by identifying the five-digit SIC codes of the relevant affected industries and proportionally assigning the relevant percentage of a given sector to be furloughed. We derive the relevant SIC codes from the Institute for Fiscal Studies in the UK [63], and we dynamically correct for any over or underestimation of furloughed workers by defining the proportion of the workforce who should be furloughed at any particular time, derived from government reports [64]. A similar dynamic correction is also applied to the key workforce. To model the more random work patterns of the remaining part of the workforce, we derive a probability that a random worker goes into the company for work from a YouGov survey [65]. We note that in many surveys, including this and others undertaken (e.g. by the ONS [66]), the methodology does not explicitly state if key or furloughed workers were included. We believe, however, that our use of these surveys presents at least a conservative estimate of work attendance.

From 20 March 2020, all schools and universities in England were asked to close, with the exception that children of key workers could still attend school. To account for the partial school reopening of Early Years (nursery and reception age children) and Year 6 students on 1 June 2020, we open up these year groups in June with an attendance compliance based on data derived from the Department for Education (DfE) [67]. While the government also asked schools to offer face-to-face support for Year 10 and 12 students from 15 June 2020, we do not include this as the sessions were generally limited and had an attendance rate around the 10% level [67]. Figure 16 shows the good agreement in the number of children attending school as derived from June compared with DfE data. The slight deviation from data after 15 June 2020, can be explained by not fully capturing the partial return of Year 10 and 12 students. The good agreement between June and the DfE data before 1 June 2020, is of particular note since this option was available only for children where all parents in the household were classified as key workers. This serves as an implicit partial validation of our method of selecting which individuals are key workers, as well as the household and company sector distribution algorithms. From 1 September 2020, we reopen schools fully in June, while accounting for a closure for the national school holidays. While the timings of this week-long holiday varies across the country, we assume all schools share the same holiday period 26 October—30 October 2020. Similarly, universities are opened from 1 September 2020, but with more restrictive social distancing measures in place. Given the modelling of where university students live, their inter-mixing is naturally captured in the household component of June (see §3.2).

On 16 March 2020, the UK Government encouraged people to avoid going to leisure venues such as bars and restaurants, although this rule was not imposed through the closure of such venues. However, on 21 March 2020, this closure took place. We model these policies first by reducing the probability that people leave the house from 16 March 2020 followed by the closure of all relevant leisure venues included in the simulation—cinemas, pubs and restaurants—from 21 March 2020. Visits to care homes are also halted from this time. Since many of these venues were permitted to reopen from 4 July 2020, we assumed all venues reopen at this point. Additionally, data collected by

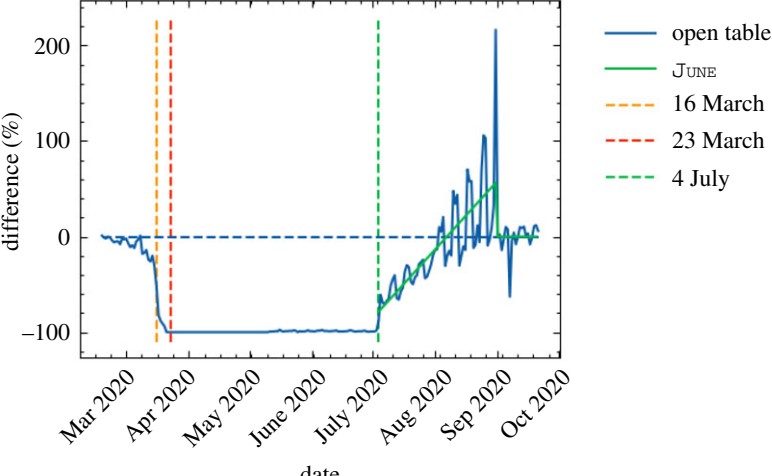

**Figure 17.** Year-on-year restaurant attendance from OpenTable [68] including a fit to the simulated reopening change in probabilities used to derive the probability that people attend restaurants in JUNE.

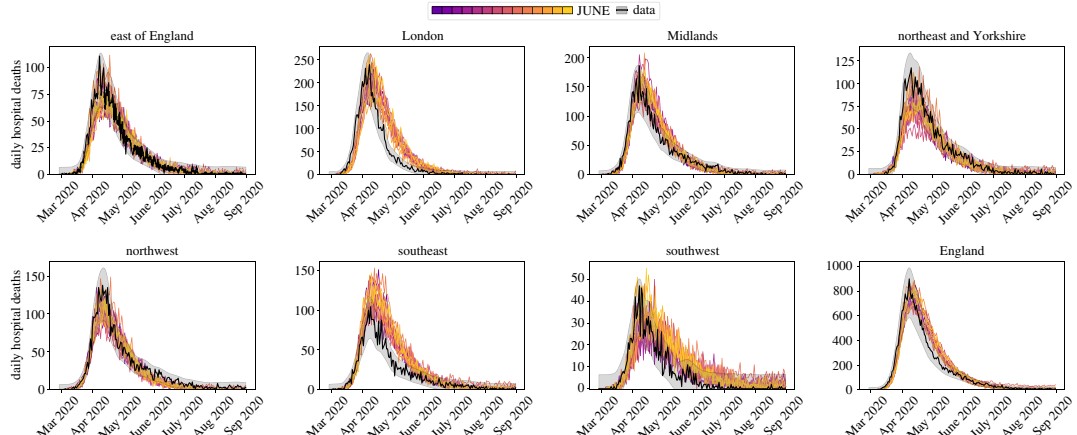

**Figure 18.** Daily hospital deaths for each region in England, and England itself, for 14 realizations of JUNE as described in this section. Each realization is illustrated as a separate colour for visibility. Observed data in black with 3 s.d. error bands. Data from CPNS [72].

OpenTable suggests that restaurant attendance after that date saw a significant increase probably encouraged by the UK Government's 'Eat Out to Help Out' scheme which we capture in JUNE (figure 17) [68,69]. For the simulation of other leisure activities, and in particular household-to-household visits, we assume a drop in compliance and a consequently increasing number of such visits. In line with data collected by the ONS [70], we model this by gradually increasing the probability of visiting another household from mid-May until 4 July 2020, when overnight visits were permitted.

# 7. Discussion of model outputs

In this section, we finally highlight the ability of JUNE to capture intricate social dynamics through a number of model outputs. It is worth noting that the realizations of JUNE presented in the following were run at parameter settings sampled from the 'non-implausible' region of the global parameter space, as defined in §8 and appendix E. See table 8 for the ranges of the global parameter space. A more complete uncertainty analysis and parameter exploration will be performed in [71].

In figure 18, we exhibit results for the number of daily deaths in hospital for regions of England and England itself. In addition, in figure 19, we show the same realizations for daily deaths in England stratified by age. The agreement with data is satisfying and while there are minor discrepancies for

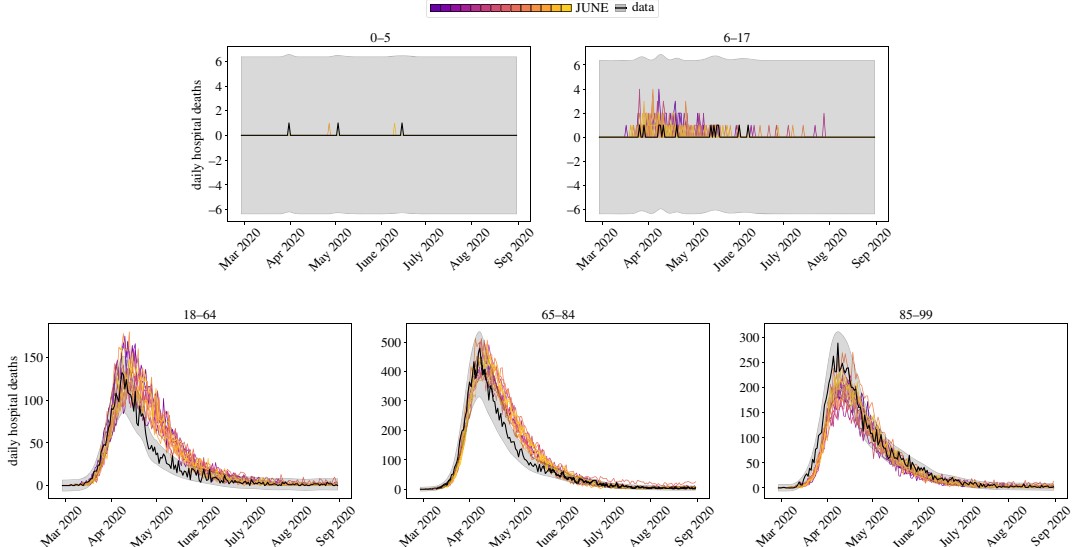

**Figure 19.** Daily hospital deaths in England stratified by age, for the same realizations as in figure 18. Observed data in black with 3 s.d. error bands. Data from CPNS [72].

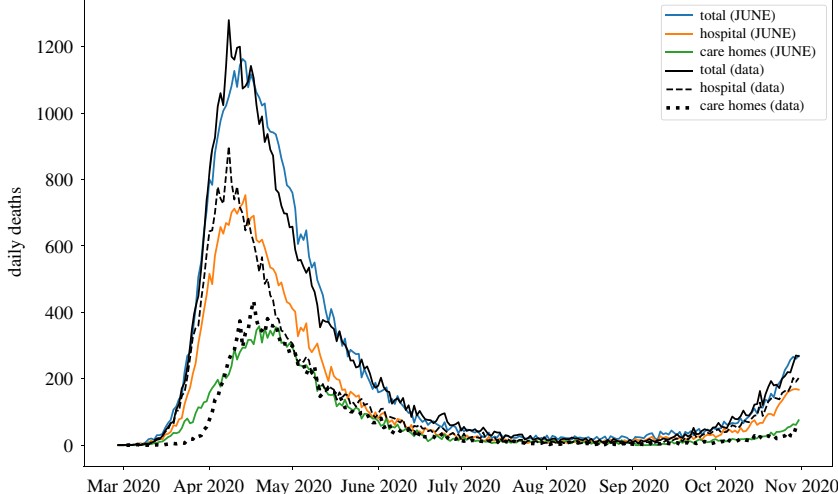

**Figure 20.** Deaths in England illustrated as different lines for total deaths, hospital deaths and deaths within care homes. Note that the total curve is the sum of hospital deaths and residence deaths (care homes as plotted, and usual households which are not plotted). Data from ONS [43].

certain outputs, we would like to stress that all of these outputs are simultaneously fit by JUNE without any region-specific parameters.

Along with deaths in hospitals, there have been a non-negligible number of fatalities in care homes in England during this pandemic. JUNE successfully models both deaths in hospitals, and deaths within care homes as illustrated in figure 20 where there is good agreement with data even into the second wave of the pandemic.

We would like to emphasize that the outputs shown here are illustrative of the capabilities of JUNE to capture the social dynamics of a heterogeneous population giving rise to large differences in disease spread to different age strata and regions.

All interactions resulting in infections are stored in full detail in the model's output, enabling further ex-post analysis of the sociological nature of disease spread and outcomes for all individuals modelled in the simulation. A simple example of such an analysis is shown in figure 21 where locations of infections are compared for one of the realizations shown in figure 18. Remaining realizations manifest a similar hierarchy of infection locations demonstrating JUNE's physical consistency across parameter space. A further, more involved, example of this type of detailed ex-post analysis can be found in [73].

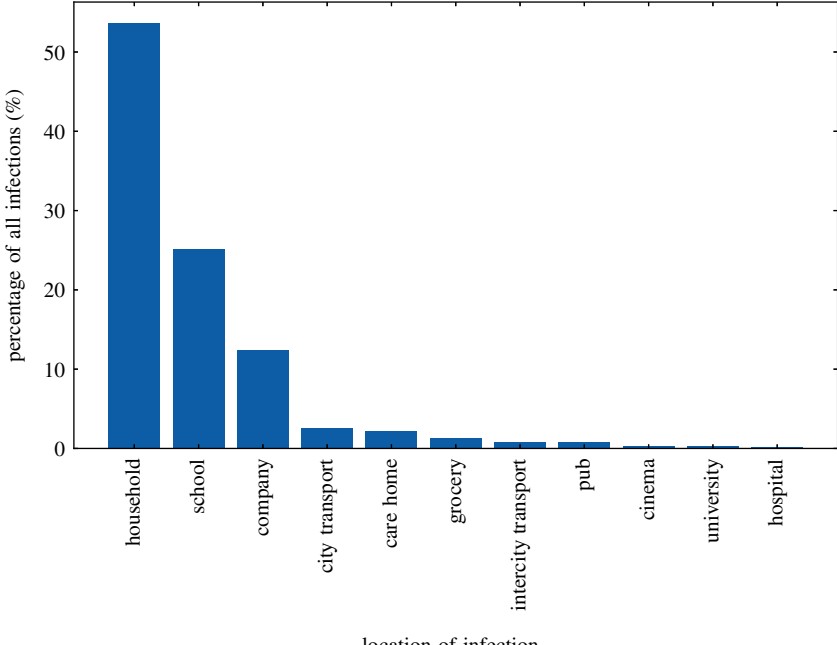

**Figure 21.** Locations where infections take place in one realization of June from figure 18. This is a simple illustrative example of the type of analysis that you can carry out using the detailed outputs of June.

# 8. Fitting via Bayesian emulation

We now discuss efficient calibration strategies which form a critical part of our ability to extract core insights from June. Fitting a complex model such as June to observed data presents a challenging task. This is mainly due to (i) the detailed nature of June and the inevitable computational expense of performing model evaluations, (ii) the large number of input parameters that we may wish to explore, (iii) the stochastic nature of the output of June, and (iv) the various uncertainties present in the comparison between model and data. A typical full England run of June like those shown in figure 18 would take approximately 10 h to complete on 64 cores (Intel Xeon Skylake) and 128 GB of memory. The combination of computational expense and high dimensional input parameter space precludes the use of many parameter exploration methods that rely upon large numbers of model evaluations (including many standard optimizers, sampling approaches such as Markov chain Monte Carlo etc.). The stochastic output, which implies we will be exploring a much more complex surface, requires methods developed to deal with stochastic functions. Even more challenging is that the substantial uncertainties present imply that we may not even want to optimize for a single 'best fit to data' as it may have limited statistical relevance, but instead search for the set of all input parameter values that give acceptable matches between model output and observed data, thereby fully capturing the induced parametric uncertainty.

We hence employ the Bayes linear emulation and history matching methodology [74–76], a widely applied uncertainty quantification approach designed to facilitate the exploration of large parameter spaces for expensive-to-evaluate models of deterministic or stochastic form. This approach centres around the concept of an *emulator*: a statistical construct that mimics the slow-to-evaluate scientific model in question, providing predictions of the model outputs with associated uncertainty, at as-yet-unevaluated input parameter settings. In contrast to the model, the emulator is extremely fast to evaluate: for example, in the case of June, the emulator exhibited a speed increase of nine orders of magnitude. The emulator provides insight into the model's structure and, thanks to its speed, can be used to perform the global parameter search far more efficiently than approaches that attempt to use the comparatively slow scientific model itself. Here we give a brief overview of emulation and history matching, but for more details see appendix E. See also [76–79] for further examples of its application within epidemiology, [80] for a comparison with approximate Bayesian computation in an epidemiological setting, and [81] for a tutorial introduction in the context of systems biology. For an extensive treatment, see [75] along with the discussion in [82]. See also [83] for a general introduction to emulation.

Initially, we identify a large set of input parameters to search over, primarily composed of interaction intensity parameters at the group level, along with associated broad ranges, as given in table 8. We then

identify a set of particular model outputs to match to corresponding observed data. Here, we focus on hospital deaths (CPNS [72]) and total deaths (ONS) at well-spaced time points throughout the period of the first wave of the epidemic. We then construct Bayes linear emulators for each of the model outputs at each of the chosen time points. The emulators are trained using a set of June runs, initially designed using a 18-dimensional Latin hypercube, and seek to mimic the behaviour of each of the June outputs as a function over the 18-dimensional parameter space. The emulators provide, at each unevaluated input location, an expectation for the possible June model output value and a position-dependent variance representing the emulator's uncertainty about this estimate. Close to known runs the emulator's uncertainty will be low; however, it will increase appropriately as we move to less well-explored regions of the parameter space [81]. Note that we deliberately choose to emulate the direct physical outputs of the model as this has multiple benefits for emulator construction, in contrast to emulating a combined metric such as the likelihood (for discussion of this point see [75,81,82]).

Due to the emulators' speed, they are ideal for global parameter exploration. This is performed by constructing an implausibility measure that gives the distance between the emulator's expected June model output and the observed data we are trying to match, standardized by all the major uncertainties present: observational errors, emulator uncertainty and structural model discrepancy, the latter being a direct acknowledgement that the model is an imperfect representation of reality (see appendix E for details). The implausibility measures are used to rule out large regions of the input parameter space that will not provide acceptable matches, and the analysis then proceeds in iterations: a second batch of June runs is performed over the remaining region of parameter space, new emulators constructed (which are only defined over this region), new implausibility measures formed and more parameter space removed. This process is referred to as iterative history matching [75,76]. See for example [78] where it was successfully applied to a stochastic disease model with 96 input parameters.

For the June model, we constructed emulators for hospital deaths and total deaths at eight time points over the period March to June, for England and for each of the seven regions, and for the age bins (defined by the SITREP dataset) 0–5, 6–17, 18–64, 65–84, 85+. The emulators were trained in three iterations formed from 125 June evaluations each. The emulators were then evaluated at 500 000 locations across the 18-dimensional input space, taking 10 min on a single processor. The results of the global parameter search are given in the optical depth plots [75] of figure 22, which shows the location of the 'non-implausible' region of interest in various two-dimensional projections of the 18-dimensional parameter space for all combinations of 12 of the most interesting input parameters (the remaining six inputs were only loosely constrained, jointly with other parameters, if at all). The June runs discussed in the preceding section were sampled from this region. Note the various joint constraints on the input parameter space imposed by the matching process, for example the strong reciprocal relationship that is required between $\beta_{school}$ and $\beta_{household}$. Similar but more complex trade-offs are identified between several other parameters, e.g. $\beta_{company}$ and $\beta_{houshold}$, $\beta_{grocery}$ and $\beta_{citytransport}$, and between $\beta_{household}$ and $M_{quarantine\ household\ compliance}$. Most parameters were not individually identifiable; however, $\beta_{company}$ and $\beta_{carevisits}$ were reasonably well constrained. For more details of this approach, of emulator diagnostics, and further output plots see appendix E.

We can see that the Bayes linear emulation and history matching methodology facilitates the efficient exploration, development and calibration of the highly complex June model using a modest number of runs, a process which would be extremely challenging to perform directly. While here we have performed a provisional exploration of the parameter space as part of the model development, for a full uncertainty analysis of the June model, including the emulator-driven generation of full probabilistic forecasts incorporating all major sources of uncertainty, see [71].

# 9. Summary

In this paper, we introduced the new June model to simulate the spread of epidemics through a population. June is an individual-based model (IBM) enabling a highly granular geographical and sociological resolution. The frequent and persisting perception that IBMs such as June are heavily parametrized and therefore lack predictive power is misleading. As noted in [84], many of the properties and building blocks of these types of model are not globally fitted to observed cases or fatalities, as is the case for deterministic and stochastic models built from differential equations. Instead, the June framework separates the uncertainty arising from unknown disease dynamics from uncertainties in the population structure, where the latter is informed by demographic statistics and other available data.

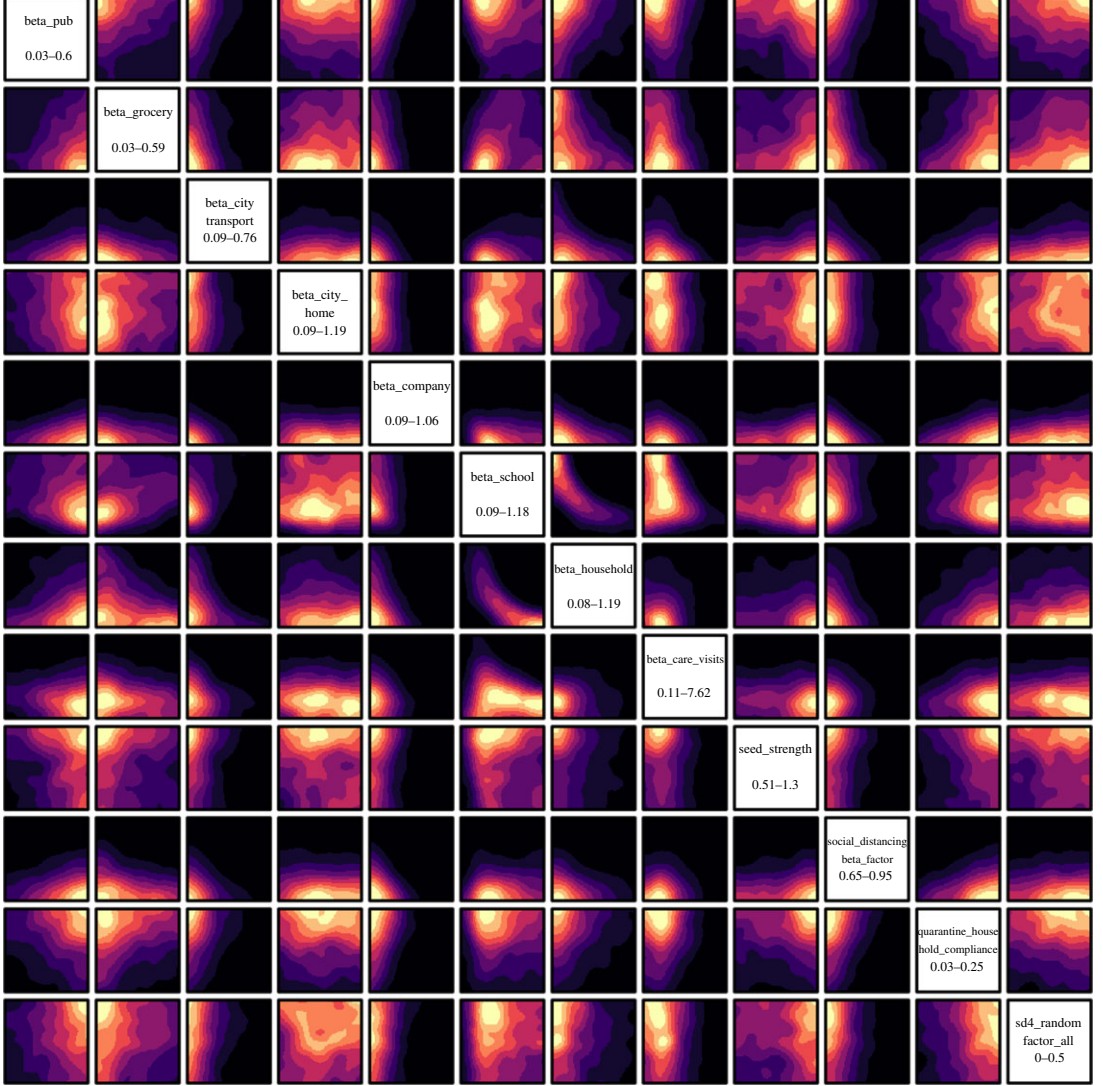

**Figure 22.** Two-dimensional projections of the 18-dimensional input space, for the 12 most interesting input parameters, coloured by the optical depth of the non-implausible region, which gives the depth or thickness of the non-implausible region conditioned on the two given inputs [81]. The ranges for each parameter are given below the parameter name in the diagonal panels. These plots are formed from 500 000 emulator evaluations over the input space. The emulators were trained on three iterations of 125 JUNE model evaluations.

The model is formulated and encoded in four distinct layers, `population`, `interaction`, `disease` and `policy`. Its modular structure allows not only the flexible and seamless addition of many details and novel features, but it also lends itself to application to other populations with different sociological set-ups. As a first example, we discuss its application to the case of the spread of COVID-19 in England, with convincing results underlining the quality of the model and its ability to understand the spread of an epidemic in great detail and with high geographical and sociological resolution.

Studies where JUNE is applied to different settings are forthcoming [85]. One of the strengths of the model is its ability to capture differences in geographical and sociological structure with unprecedented resolution, facilitated through the hierarchical structure in which the `population` is organized. JUNE also allows a flexible yet detailed modelling of daily activities of the virtual population, by combining the geographical position of buildings and other structures with the social interactions taking place. In contrast to other models this enables a very granular understanding of work patterns, leisure activities, etc. In forthcoming publications, we will exploit this high level of detail to try and answer pertinent questions relating to social imbalances in the impact of COVID-19.

Data accessibility. Map data copyrighted OpenStreetMap contributors and available from https://www.openstreetmap.org.

**Authors' contributions.** J.A.-B., A.C., C.C.-L., E.E., M.I.-L., A.Q.-B., A.S. and H.T. thank the STFC-funded Centre for Doctoral Training in Data-Intensive Science[9] for financial support.

**Competing interests.** We declare we have no competing interests.

**Funding.** F.K. gratefully acknowledges funding as Royal Society Wolfson Research fellow. I.V. gratefully acknowledges Wellcome funding (218261/Z/19/Z). This work used the DiRAC@Durham facility managed by the Institute for Computational Cosmology on behalf of the STFC DiRAC HPC Facility (www.dirac.ac.uk). The equipment was funded by BEIS capital funding via STFC capital grants ST/K00042X/1, ST/P002293/1, ST/R002371/1 and ST/S002502/1, Durham University and STFC operations grant ST/R000832/1. DiRAC is part of the National e-Infrastructure.

**Acknowledgements.** This work was undertaken as a contribution to the Rapid Assistance in Modelling the Pandemic (RAMP) initiative, coordinated by the Royal Society. We are indebted to a number of people who shared their insights into various aspects of the project with us: We would like to thank Sinclair Sutherland for his patience and support in using the ONS database of the census data—without his help it would have been near impossible for us to produce our virtual population. James Nightingale and Richard Hayes provided valuable insights into the construction of efficient algorithms in the initial phase of the project. We are grateful to Bryan Lawrence, Grenville Lister, Sadie Bartholomew and Valeriu Predoi from the National Centre of Atmospheric Science and the University of Reading for assistance in improving the computational performance of the model. We gratefully acknowledge the generous provision of computing time on the Hartree and JASMIN facilities. We would like to thank the GridPP team at Durham and Manchester for their support and computing time spent on their systems. We would also thank Michael Goldstein and T.J. McKinley for their statistical and epidemiological advice. Christina Pagel and Rebecca Shipley provided invaluable advice in producing this publication and looking for holes in our arguments. This paper made use of Python [86] and the following Python libraries: Matplotlib [87], Numpy [88], Pandas [89,90], Scipy [91], SciencePlots [92].

# Appendix A. Algorithms

## A.1. Constructing credible households

The ONS divides households into the following broad categories: single, couple, family, student, communal and other [20]. We populate the households in this ordering, giving preference to those types for which we have the most precise and unambiguous data.

We define and construct households types as follows:

1. Single: These are households with a single person living in them. The census data differentiate single households occupied by an adult or an older adult (greater than or equal to 65 years old), and we fill the households accordingly.
2. Couple: These are households occupied by a couple without children. Again, the census differentiates between household with adults or older adults living in them. We preferentially fill these households with two people of different sex, with an age difference sampled from the corresponding UK distribution of age differences at the time of marriage [93] (see also figure 23a).
3. Family: These households are defined by the number of adults (singles or couples) and the number of children. A difficulty here is that the census data does not stratify beyond 'two or more' children. To compensate for this, we introduce a distribution to select the number of children in these households. To fill a family household, we allocate a female adult first. If there are no female adults available (because they have already been allocated somewhere else), we chose a male adult. In case of families with two adults, we match the person with a partner, preferentially with different sex, and an age difference sampled from the same dataset we use for couples. The census data provide us with the number of dependent children for each OA (area), and we add a suitable number of children according to the age difference between the mother and the $n$-th child as given by ONS data collected on birth characteristics [94] (see also figure 23b,c).
4. Students: From the census data, we know how many student households there are and how many students live in a given OA (area). We uniformly distribute students among their households, assuming a constant ratio of the number of students per household. Students are selected from the population aged between 18 and 25 years old.
5. Communal: We use census data on the number of people in an OA (area) living in a communal establishment, as well as the number of such establishments, such as care homes [21]. The communal establishments are filled last, after the types described above; their residents will be those who do not live in any of the other household types. As in the case of student households, we assume a constant ratio of the number of communal residents per establishment.

---

[9]https://ddis.physics.dur.ac.uk/.

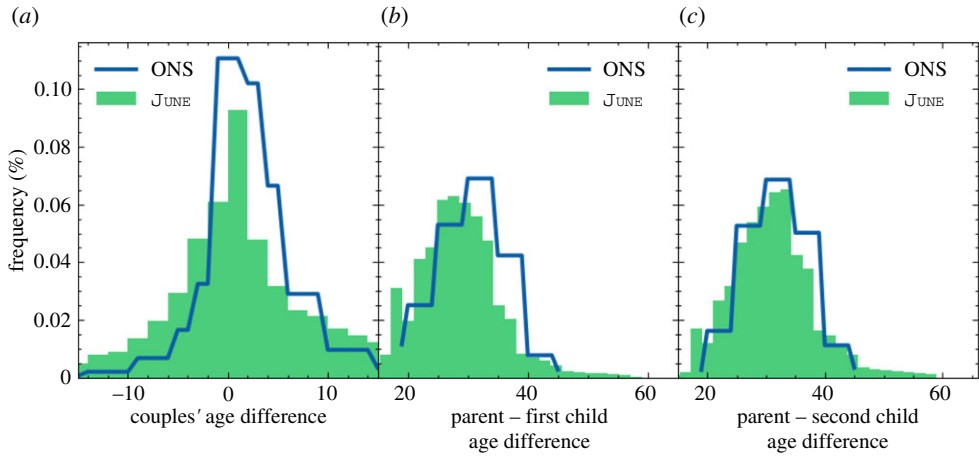

**Figure 23.** Distributions of age differences between partners (*a*), between parents and their first (*b*) and second child (c): outputs of JUNE compared with the input data from the ONS database.

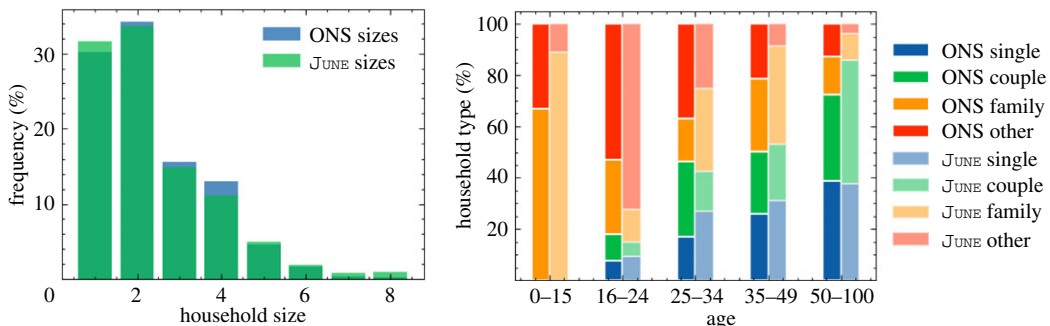

**Figure 24.** Comparisons between outputs of JUNE and data from the ONS database for all England. (*a*) Household sizes, (*b*) household composition by age.

6. Other: This category encapsulates the uncertain household compositions given by ONS. These may include groups of adults living together, multi-family or multi-generational families. In a similar manner to the communal households, these are filled last with those people that have not yet been allocated.

As a further test of our household populating algorithm against available data, we compare the JUNE household size distribution and age dependence of people living in different household types with that given by ONS [20]. Figure 24 demonstrates that the JUNE household composition algorithm clearly produces a household size distribution in good agreement with the census data. We also observe the impact of our assumptions on the composition of families and more complex household compositions (other). Given the unknown specifics of certain household composition types, we believe our overall household composition by age to be in reasonable agreement with the data.

## A.2. Schools

The procedure for assigning children and teachers to schools throughout England is specified in §3.3.

Following our algorithm, we arrive at a distribution of school sizes displayed in figure 25, which we see to be in reasonable agreement with the data. Similarly, figure 26 shows the full distribution of class sizes in JUNE. In the case of COVID-19, most countries have prioritized the return of children to school from younger age brackets. Therefore, recovering good agreement with data particularly in these age brackets is crucial.

## A.3. Workplaces

We use ONS data on industries and companies in England categorized according to 21 sectors following the Standard Industrial Classification (SIC) code convention (table 1) [26] as our framework for differentiating between different types of work.

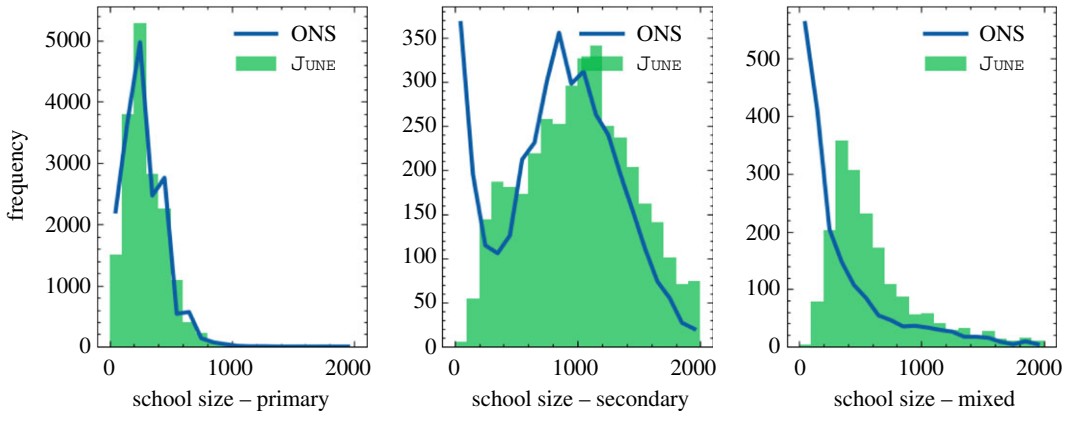

**Figure 25.** Distribution of school sizes comparing the JUNE simulation with data.

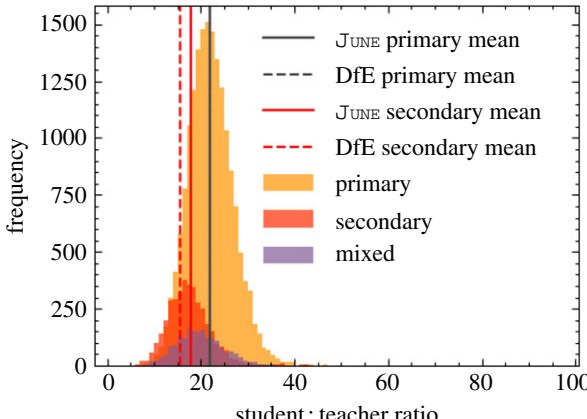

**Figure 26.** Distribution of student to teacher ratios for primary schools, secondary schools and mixed schools.

Companies are initialized according to ONS data on company sizes and sectors at the MSOA (super area) level [27]. We use data on the geographical distribution of company sizes to fix the number of companies at the MSOA (super area) level and use the data on the distribution of sectors to probabilistically assign an industry sector to these companies at the same geographical level. Since the ONS provides information on company sizes by binned size ranges, we take the median size of each bin and assign this to each company. The largest bin is 1000 + employees which we assume to be 1500. It should be noted that companies are not assigned a sector based on their size, but purely on their geography. This does not mean there is no correlation between company size and their sector in JUNE, but that this would arise implicitly based on the geographical distributions, rather than explicitly from data input.

Individuals are assigned a sector attribute probabilistically, following the distributions of sectors dis-aggregated by sex at the MSOA (super area) level [28]. We determine the MSOA (super area) in which they work according to the ONS commuting origin–destination matrix (or 'flow' data) [29] which provides information on the number of people by sex travelling from one MSOA (super area) to another for work. Finally, a matching is carried out between people who work in a certain MSOA (super area), and the companies available to them based on their respective sector attributes. In future work, we plan to use additional demographic attributes to assign individuals their sectors and companies.

## A.4. Commuting

The commuting structure in JUNE is built upon the national transport network constructed from nodes representing cities, and edges representing possible transit routes. Commuters are defined as either 'internal', i.e. they live and work in a city, or 'external', i.e. they live outside a city's metropolitan boundary and commute into it (see §3.4 for more details on how people are assigned locations of work). The metropolitan boundary of each city is defined using data collected by the ONS [36] which specifies the MSOAs (super areas) that belong to the cities.

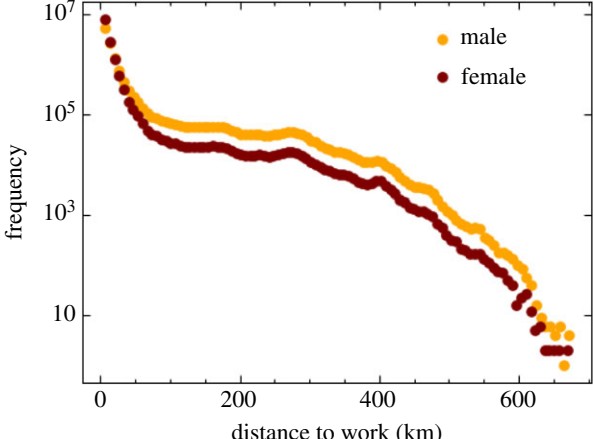

**Figure 27.** Distance travelled to work by sex according to June. Here we see that men are more likely to travel further to work then women. This is in broad agreement with data presented in [30].

The following procedure is used to determine the groups within which people have the chance to mix during a commute.

1. For each city, we seed several additional nodes which act as 'gateway stations' outside the metropolitan area boundary. These serve as funnels into the city and determine the mixing of external commuters. In the case of London, we seed eight stations which are placed evenly around the boundary of the metropolitan area. For all other stations, we seed four evenly spaced stations north, south, east and west of the city boundaries. These figures are informed by the approximate number of train lines entering each city, and the proportional differences between the number of London public transport links and those of other cities [37].

2. We model the commuting of all people who travel by public transport into a city's metropolitan area. We assign all external commuters to the nearest gateway station to where they live. During each commuting time-step in the simulation, people travelling through the same gateway station are randomly split into 'carriages' containing people with whom they have the potential to interact. Similarly, internal commuters are also split into carriages and able to interact with each other.

3. During a commute time-step, each carriage is assigned to be travelling at 'peak time' with an 80% probability.

4. The default number of people in an average carriage is fixed to 50 people. For each city this number is adjusted in proportion to data from the UK Department for Transport (DfT) data on overcrowding in trains [34]. This data also disaggregates at the level of peak or off-peak travel which is used to further adjust the filling of carriages.

5. The commuting time-step is run twice a day in order to simulate commuting in each direction.

We calculate the distance travelled to work by sex, in figure 27, and we see that men are more likely to travel further to work in our model than women. Our findings are in reasonably good agreement with the survey [30] and serve as an independent validation of our model.

# Appendix B. Time-steps

As mentioned in §4.1, June time-steps allow differentiation between weekdays and weekends, and have a number of allotted activities. The default time-steps are described in tables 5 and 6.

When choosing the time-steps, we aimed to choose the lengths such that they are somewhat close to the characteristic time of interaction of activities allowed in that time-step, but also not choosing so many time-steps to overfragment the simulation. For instance, the weekday time-step with index 1 (09.00–17.00) is 8 h, and matches the primary activities of 'school' and 'work', even though the 'leisure' activity (which is allowed for old adults who are not assigned a workplace) has a characteristic time of 3 h. Breaking this in half would better match the leisure characteristic time (3 h) for this time-step, but would mean that all individuals in the simulation would be reassigned an

**Table 5.** Time-steps and allowed activities for a calendar weekday, where M, R, C, P, L are 'medical facility', 'residence', 'commute', 'primary activity' and 'leisure', respectively.

| index | calendar time | allowed activities |
|---|---|---|
| 0 | 08.00–09.00 | M, R, C |
| 1 | 09.00–17.00 | M, P, L, R |
| 2 | 17.00–18.00 | M, R, C |
| 3 | 18.00–21.00 | M, L, R |
| 4 | 21.00–08.00 | M, R |

**Table 6.** Time-steps and allowed activities for a calendar weekend. Allowed activities are the same as in table 5.

| index | calendar time | allowed activities |
|---|---|---|
| 0 | 08.00–12.00 | M, R, L |
| 1 | 12.00–16.00 | M, R, L |
| 2 | 16.00–20.00 | M, R, L |
| 3 | 20.00–08.00 | M, R |

activity for the second of the two time-steps. Even though the vast majority would be reassigned to their same, required 'primary activity', causing needless computation.

# Appendix C. Contact matrices

We use the contact matrices from the BBC Pandemic survey [15] and supplement them with the `PolyMod` matrices [10] for interactions of children with other children in the age bracket of 0–12 years. When comparing the matrices that capture interactions in all settings given in the BBC study, an anomaly appears in the matrix describing physical contacts—the original `PolyMod` data approximately a factor of 3 higher than the BBC matrices in neighbouring age bins. We account for that by a simple scaling of the physical contacts by 1/3 before using these data. To arrive at matrices including interactions at home, in school, or in other settings for the age brackets 5–12, missing in the BBC study, we scale the `PolyMod` setting-inclusive results by a ratio of contacts in the respective setting for the age bin of 13–14-year-old kids, while we assume that the interactions of 0–4-year-old children are concentrated at home.

To extract mixing matrices that are suitable for our context-specific simulation, we have to correct for the fact that the reported matrices average over the corresponding age bins in the UK population. For example, contacts between teachers and school children are normalized to the full UK population in the respective age bin instead of the number of teachers in schools that actually participate in the interaction. This necessitates rescaling to the number of people in the social context to arrive at corrected social interaction matrices $\bar{\mathcal{M}}_{ij}^{(H,W,S,O)}$. This correction step will be detailed in the relevant subsections below.

## C.1. Social mixing at work

The matrices for the age-dependent interaction frequency at the workplace show only a very mild correlation with age, typically favouring interactions of workers with a similar age by about a factor of 2. We will therefore not include age effects at the workplace into the matrices used in June. To minimize effects due to early retirement, students etc. we average over the ages of 25–60 and we compare this to the average over the working age, 18–64, but correct for an employment rate of 75%. In so doing, we arrive at the number of daily contacts for adults at work:

$$n_{AA}^{(W)} = \begin{cases} 4 & (0.35 \text{ physical}) \text{ for ages } 25-60 \\ 4.8 & (0.35 \text{ physical}) \text{ for ages } 18-64, \text{ corrected for employment rate.} \end{cases} \quad \text{(C 1)}$$

In June, we will use $n_{AA}^{(W)} = 4.8$, with a ratio of about 7% physical contacts. While it is obvious that different industrial sectors will in reality have very different numbers of daily contacts, with corresponding impact on their vulnerability towards infections, we have not made any attempt to account for such a sector-dependent modulation, apart from effects that naturally arise from different sizes of workforces in different companies.

## C.2. Social mixing in schools

We decompose school populations into year groups labelled with indices $i \in \{1, 2, \ldots, N\}$ for a school with $N$ year groups and denote teachers with $T$. Starting with the interaction of pupils in various year groups an apparent large asymmetry emerges between the summed number of interactions of pupils with adults in the school and of adults with pupils in the BBC dataset. This, however, is easily explained by realizing that the number of interaction in a given context is normalized to the fraction of the population in a given age bin, irrespective of whether they can participate in the interaction or not. This means that the number of interactions between teachers and pupils have to be renormalized to the ratio of teachers in the adult population—about 500 000 teachers out of 36 300 000 adults, with about 216 000 working in primary and 208 000 working in secondary schools.

Summing the number of interactions of children in the age range of 5–17 with adults in the range 25–65 in schools, and assuming the latter are all teachers yields an average of 0.75 pupil–teacher interactions (0.06 = 8% of them physical) per day with very little dependence on the children's age. Conversely, adults have about 0.2 (0.02 = 10% of them physical) interactions per day with children in schools, again, relatively independent of the age of the children. Normalizing this to the number of teachers in the population, we arrive at about 15 teacher–pupil interactions per day, which fits very well to approximate teacher–pupil ratios of $1:20$–$1:25$.[10] We therefore assume that the individual interaction frequency of one specific teacher–pupil pair is consistently described with 0.75/day. For interactions among adults in the school setting, we include the interaction of parents with teachers and of parents among themselves, thereby blurring the picture. We therefore assume that teachers inherit the daily contact frequencies from the workplace mixing above. Turning finally to the interactions amongst children, we see a very dominant correlation in age. In order to capture this, we assume that per year of age-difference the number of interactions among children in school, $n_{KK}^{(S)}$, will be reduced by a factor $\xi$. By fitting to the combination of BBC and `PolyMod` studies we find, to good approximation, $x_i = 0.3$ and $n_{KK}^{(S)} = 2.5$, with on average 15% of the interactions being physical.

As a consequence, we obtain the following social interaction frequency matrix for individual pairings at schools:

$$\bar{\mathcal{M}}_{ij}^{(S)} \approx \begin{pmatrix} 4.8 & 0.75 & 0.75 & 0.75 & \ldots \\ 15 & 2.50 & 0.75 & 0.25 & \ldots \\ 15 & 0.75 & 2.50 & 0.75 & \ldots \\ 15 & 0.25 & 0.75 & 2.50 & \ldots \\ \vdots & \vdots & \vdots & \vdots & \ddots \end{pmatrix}, \tag{C 2}$$

where the first row and the first column specify the interactions between teachers and students in different year groups, and the second and following row and columns are populated by interactions of the pupils with other pupils across year groups ordered by age.

## C.3. Social mixing at home

In our model, we decompose the household population into four subgroups, namely children ($K$, ages 0–19), young adults ($Y$, 18–24), adults ($A$, 25–64) and older adults ($O$, 65+). We therefore arrive at a $4 \times 4$ matrix of corrected social interactions at home, $\bar{\mathcal{M}}_{ij}^{(H)}$, where the indices $i, j \in \{K, Y, A, O\}$. In the following, we will detail how we arrive at the various matrix elements. When correcting for the impact of social environment, i.e. the household compositions, we will ignore household compositions which are listed as 'other' in the ONS database, due to a lack of detailed information (see §3.2 for more details). When using these data, we will use numbers in units of millions, $H_{OAYK}$ of households with a composition of $O$ older adults, $A$ adults, $Y$ independent children or young adults living at home and $K$ children aged 0–19.

---

[10]In fact, for primary schools, the average class size is about 21 pupils, while for secondary schools it is about 16 pupils [25].

— $\bar{\mathcal{M}}_{OO}^{(H)}$: we ignore the case of care homes or other facilities with more than two residents. Then the average interaction frequency from the BBC data is given by $n_{OO}^{(H)} = 0.78$ (0.44 physical) and 0.62 at weekends.[11] With $H_{2000} = 2.131$ and $H_{1000} = 3.294$.[12]

$$\bar{\mathcal{M}}_{OO}^{(H)} = 0.78 \times \frac{2H_{2000} + H_{1000}}{2H_{2000}} \approx 1.4. \tag{C3}$$

— $\bar{\mathcal{M}}_{AA}^{(H)}$: the interaction frequency between adults aged 20–65 at home from the BBC data is given by $n_{AA}^{(H)} = 1.2$ (0.74 = 62% of them physical).

$$\bar{\mathcal{M}}_{AA}^{(H)} = 1.2 \times \frac{\sum_{x,y} (2H_{02xy} + H_{01xy})}{\sum_{x,y} 2H_{02xy}} \approx 1.34, \tag{C4}$$

where $\sum_{x,y} H_{02xy} = 8.751$ and $\sum_{x,y} H_{01xy} = 7.644$.

— $\bar{\mathcal{M}}_{YY}^{(H)}$: the interaction frequency between young adults age 18–26 at home from the BBC data is given by $n_{YY}^{(H)} = 1.3$ (0.4 = 34% of them physical). There is no obvious household correction that we can apply, but the number of contacts is relatively close to the value of $\bar{\mathcal{M}}_{AA}^{(H)} = 1.34$, so we will assume that young adults interact with each other with a frequency similar to that of adults

$$\bar{\mathcal{M}}_{YY}^{(H)} = \bar{\mathcal{M}}_{AA}^{(H)}. \tag{C5}$$

It is worth noting that the age range for young adults is relatively narrow, and that there will be edge effects that may effectively increase the interaction frequency.

— $\bar{\mathcal{M}}_{YA}^{(H)}$ and $\bar{\mathcal{M}}_{AY}^{(H)}$: we have $n_{YA}^{(H)} \approx 0.7$ with a relatively steep decline with the age of the young adults, which we attribute to the fact that with increasing age young adults move out of their parents' home. To obtain some better understanding of the situation, we look at the interaction of adults in the age range 40–65 with young adults, aged 18–24. From this, we arrive at an average of $n_{AY}^{(H)} = 0.17$ (0.07 = 40% of them physical).

To relate this to a corrected value, we must make an assumption concerning the number of young adults in the three age bins that still live with their parents, which we take as 75%, 50% and 40% for the three age bins. To correct the $AY$ number, we assume that the majority of households with young adults living as non-dependent children with their parents is composed of households with one young adult. Therefore,

$$\left.\begin{aligned} \bar{\mathcal{M}}_{YA}^{(H)} &= \frac{1}{3}\left[\frac{0.87}{0.75} + \frac{0.65}{0.5} + \frac{0.55}{0.4}\right] \approx 1.3 \\ \text{and} \quad \bar{\mathcal{M}}_{AY}^{(H)} &= 0.17 \cdot \frac{\sum_{xy} (2H_{02xy} + H_{01xy})}{\sum_{y} (2H_{021y} + H_{011y})} \approx 1.47, \end{aligned}\right\} \tag{C6}$$

where $\sum_{y} H_{021y} = 1.514$ and $\sum_{y} H_{011y} = 0.946$.

— $\bar{\mathcal{M}}_{KK}^{(H)}$: the average number of daily contacts at home between children age 0–17 is $n_{KK}^{(H)} = 0.47$ (79% of them physical). Assuming all children live as dependents with their parents, and demanding that households with '2 or more children' (ONS classification) have, on average, 2.3 children to account for the UK reproduction rate, we arrive at

$$\bar{\mathcal{M}}_{AA}^{(H)} = 0.87 \cdot \frac{\sum_{x} (H_{02x1} + H_{01x1}) + 2.3(H_{02x2} + H_{0.1x2})}{\sum_{x} 2.3(H_{02x2} + H_{0.1x2})} \approx 1.2. \tag{C7}$$

— $\bar{\mathcal{M}}_{KA}^{(H)}$ and $\bar{\mathcal{M}}_{AK}^{(H)}$: to account for contacts of children with adults we will use sliding age windows in dependence on the age of the child, using that parents are usually between 20 and 40 years older than their children. We then arrive at $n_{KA}^{(H)} = 1.27$ (70% of them physical) and $n_{AK}^{(H)} = 0.67$, the former with an only mild dependence on the age of the child, while the latter shows clear edge effects for the first and last bins of the adult age distribution. These numbers translate into

$$\bar{\mathcal{M}}_{KA}^{(H)} = 1.27 \tag{C8}$$

[11]One may speculate in how far this drop is a reflection of uncertainties in the data or a true 'physical' effect, for example due to visitors, travel or similar.

[12]Here and in the following, the numbers of different household configurations are taken from [95].

and

$$\bar{\mathcal{M}}_{AK}^{(H)} = 0.67 \cdot \frac{\sum_{x,y}(2H_{02xy} + H_{01xy})}{\sum_{[]x,y}2(H_{02x1} + H_{02x2}) + (H_{01x1} + H_{01x2})]} \approx 1.69.^{13} \tag{C 9}$$

We will also assume that the interaction frequency and intensity of children and young adults living in the same household is determined by

$$\bar{\mathcal{M}}_{KY}^{(H)} = \bar{\mathcal{M}}_{KA}^{(H)} \quad \text{and} \quad \bar{\mathcal{M}}_{YK}^{(H)} = \bar{\mathcal{M}}_{AK}^{(H)}. \tag{C 10}$$

— $\bar{\mathcal{M}}_{O,KYA}^{(H)}$ and $\bar{\mathcal{M}}_{KYA,O}^{(H)}$: we assume that interactions of children, young adults and adults with older adults at home have three different realizations:
   1. as regular contacts in a multi-generational household, where we assume that older adults behave like adults in terms of interaction frequency and intensity;
   2. as regular contacts between children and their grandparents who act as child-minders while the parents are at work;
   3. through regular or sporadic visits, where we again assume that interactions of older adults follow the pattern of adults.

As a result, we obtain the following social mixing matrix

$$\bar{\mathcal{M}}_{ij}^{(H)} = \begin{pmatrix} 1.2 & 1.69 & 1.69 & 1.69 \\ 1.27 & 1.34 & 1.47 & 1.50 \\ 1.27 & 1.30 & 1.34 & 1.34 \\ 1.27 & 1.50 & 1.34 & 2.00 \end{pmatrix} = \begin{array}{c|c|c|c|c} & K & Y & A & O \\ \hline K & 1.2 & 1.69 & 1.69 & 1.69 \\ Y & 1.27 & 1.34 & 1.47 & 1.50 \\ A & 1.27 & 1.30 & 1.34 & 1.34 \\ O & 1.27 & 1.50 & 1.34 & 2.00 \end{array}, \tag{C 11}$$

where, for convenience, we have made the entries explicit.

## C.4. Social mixing in other venues

Social venues (pubs, cinemas and groceries) in JUNE are assumed to have only one subgroup, 'attendees', meaning that the social mixing matrix for these interactions is a single-element; $\bar{\mathcal{M}}^{(P)} = 3$, $\bar{\mathcal{M}}^{(C)} = 3$, $\bar{\mathcal{M}}^{(G)} = 1.5$ for 'pubs', 'cinemas' and 'groceries', respectively. These values were chosen heuristically according the estimated number of contacts in each location relative to the the number of contacts set elsewhere (as discussed above). Given that we do not consider different subgroups in these locations, making the matrix single-valued, these numbers only serve the purpose of intuitively introducing a hierarchy of contact intensities ($\beta$ parameters) into the model structure. Since the intensity parameters are fitted to data (see §8), the form of equation (5.2) ensures that the choice of these social mixing matrices values will not significantly affect the probabilities of transmission.

Hospitals have three subgroups: medical staff, ward patients and ICU/ITU patients. The social mixing matrix for hospitals (where the superscript M refers to 'medical facility') is

$$\bar{\mathcal{M}}_{ij}^{(M)} = \begin{pmatrix} 5 & 10 & 10 \\ 1 & 0 & 0 \\ 1 & 0 & 0 \end{pmatrix} \quad \text{and} \quad \phi_{ij}^{(M)} = \begin{pmatrix} 0.05 & 1 & 1 \\ 1 & 0 & 0 \\ 1 & 0 & 0 \end{pmatrix}, \tag{C 12}$$

where $(i, j) \in \{S, W, I\}$ denoting the three subgroups, medical staff, ward patients and ICU/ITU patients, respectively. The number of contacts between a medic and patients, 10, represents the average number of patients per medic. We assume that a patient is visited by a medic once per characteristic time, set to 8 h for hospitals. The number of contacts between patients is irrelevant, as patients are by definition already infected, but is set to zero.

Social mixing in care homes considers three subgroups: workers, residents and visitors, with matrix

$$\bar{\mathcal{M}}_{ij}^{(CH)} = \begin{pmatrix} 15 & 15 & 1 \\ 1.5 & 4 & 20 \\ 1.5 & 6 & 0 \end{pmatrix} \quad \text{and} \quad \phi_{ij}^{(CH)} = \begin{pmatrix} 0.05 & 1 & 0 \\ 1 & 0.4 & 1 \\ 0 & 0.5 & 0 \end{pmatrix}, \tag{C 13}$$

with a characteristic time of 24 h. The seemingly high number of contacts between workers and visitors, and residents and visitors is to compensate for the characteristic time of 24 h; if visitors were to be present

---

[13]Note that the latter number becomes 2.5 if we only use the central bins of parent ages 35–50 and the corresponding number of contacts $n_{AK}^{(H)} = 0.96$, which, however, introduces a bias in favour of the more intense interactions with children in primary school age.

**Table 7.** Characteristic functions and their parameters.

| name | function | source |
|---|---|---|
| $C_T$ | constant with time $T$ | |
| $\beta_I$ | $\beta_{2.29,19.05,0.39,39.8}(t)$ | [40] |
| $LN_M$ | $LN_{0.83,5.7}(t)$ | * |
| $\beta_H$ | $\beta_{1.35,3.68,0.05,27.1}(t)$ | [96] |
| $\beta_D$ | $\beta_{1.21,1.97,0.08,12.9}(t)$ | [96] |
| $LN_{ICU}$ | $LN_{1.41,0.9}(t)$ | [97] |
| $e_{ICU}$ | $e_{1.06,0.89,12}(t)$ | [97] |
| $e_D$ | $e_{1.23,1,9.69}(t)$ | [97] |

*We constrained the time from symptom onset to hospitalization through private communication with hospital physicians at early stages of the first COVID-19 wave of infections. We later checked this assumed profile against published data and found our values to be broadly consistent.

in a care home for a full characteristic time, they would experience this many contacts, but visits are a daytime activity which takes only a few hours, resulting in fewer contacts.

Finally, universities are modelled as having six groups to represent professors and five distinct groups of students (for the moment based only on age 19–23), with diagonal elements $\bar{\mathcal{M}}_{i=j}^{(U)} = 2$ and off-diagonal elements $\bar{\mathcal{M}}_{i\neq j}^{(U)} = 0.75$, and all $\phi_{ij}^{(U)} = 0.25$.

## C.5. Deriving contact matrices from JUNE

We derive the contact matrices in figure 9 by simulating a week of pre-lockdown activity. For each person, in each subgroup $i$, in each venue, we choose the required $N_{ij}$ people (with replacement) for all (non-empty) subgroups $j$ in that venue (where $N_{ij}$ is from the relevant social mixing matrix). We populate 'raw' contact matrices using these selected people. As these contacts are then uni-directional, we make the same corrections as in [15] to account for reciprocal contacts. We hope to produce contact matrices derived from constructing self-consistent (reciprocal) networks of contacts within groups in future work.

# Appendix D. Details on modelling health trajectories

For the times spent in different stages of disease progression, we use a variety of functions, namely intervals of constant length, scaled and shifted $\beta$ functions, scaled log-normal distributions and exponential Weibull distributions, given by

$$\left.\begin{aligned} C_{t_{end}}(t) &= \Theta(t_{end} - t), \\ \beta_{a,b,l,S}(t) &= \beta_{a,b}\left(\frac{t-l}{S}\right), \\ LN_{s,S}(t) &= LN_s\left(\frac{t}{S}\right) \\ \text{and} \quad e_{a,c,S}(t) &= e_{a,c}\left(\frac{t}{S}\right). \end{aligned}\right\} \tag{D 1}$$

The trajectories and their building blocks to construct the corresponding time intervals infected individuals spend in various stages of the disease are listed in table 3 (table 7).

# Appendix E. Calibration via Bayes linear emulation and history matching

We now provide more details of the Bayes linear emulation and history matching process outlined in §8. To set up the history matching problem, we identify a large set of 18 input parameters to the JUNE model to explore. This set is composed mainly of contact intensity parameters, but also contains such

**Table 8.** The input parameters that form the 18-dimensional vector $x$ explored in the global parameter search, their type and their ranges that define the search region $\mathcal{X}_0$. The parameters $\alpha_{\text{seed strength}}$ and $M_{\text{sd4 random factor all}}$ modulate the strength of the seeding process, and the social distancing policy active from 7 July, respectively.

| input parameter ($x_i$) | type | range |
|---|---|---|
| $\beta_{\text{pub}}$ | contact intensity | [0.02,0.6] |
| $\beta_{\text{grocery}}$ | . | [0.02,0.6] |
| $\beta_{\text{cinema}}$ | . | [0.02,0.6] |
| $\beta_{\text{university}}$ | . | [0.02,0.6] |
| $\beta_{\text{city transport}}$ | . | [0.08,0.77] |
| $\beta_{\text{intercity transport}}$ | . | [0.08,1.2] |
| $\beta_{\text{hospital}}$ | . | [0.08,1.2] |
| $\beta_{\text{care home}}$ | . | [0.08,1.2] |
| $\beta_{\text{company}}$ | . | [0.08,1.2] |
| $\beta_{\text{school}}$ | . | [0.08,1.2] |
| $\beta_{\text{household}}$ | . | [0.08,1.2] |
| $\beta_{\text{care visits}}$ | . | [0.1,8] |
| $\beta_{\text{household visits}}$ | . | [0.08,1.2] |
| $\alpha_{\text{physical}}$ | physical contact factor | [1.8,3] |
| $\alpha_{\text{seed strength}}$ | seeding | [0.5,1.3] |
| $M_{\text{quarantine household compliance}}$ | compliance | [0.034,0.26] |
| $M_{\text{social distancing beta factor}}$ | social distancing | [0.65,0.95] |
| $M_{\text{sd4 random factor all}}$ | social distancing | [0.004,0.5] |

parameters governing social distancing effects, compliance and physical contact, with each parameter specified along with associated broad ranges, in table 8. We denote this set of parameters by the 18-dimensional vector $x$ and denote the initial search region defined by their combined ranges as $\mathcal{X}_0$. We identify a set of outputs to match to observed data, specifically the deaths and total deaths for England and for each of the seven regions, and for the age bins 0–5, 6–17, 18–64, 65–84, 85+, at the time points of 20 March, 28 March, 5 April, 13 April, 21 April, 29 April, 12 May, 26 May, 8 June, 8 June and 23 June 2020. We represent the list of all these outputs as the vector $f$.

We note that the June model can now be viewed as a function that maps the inputs $x$ to the vector of all outputs of interest $f(x)$. As we cannot evaluate the model $f(x)$ exhaustively over the full parameter space $\mathcal{X}_0$ due to computational expense, we mimic it using a fast-to-evaluate (but uncertain) Bayesian emulator. For an individual output $f_i(x)$, representing for example, the total deaths in England on 28 March, we construct an emulator of the form

$$f_i(x) = \sum_j b_{ij} g_{ij}(x_{A_i}) + u_i(x_{A_i}) + v_i(x), \qquad (E\,1)$$

where the first term on the r.h.s. is a regression term designed to capture global behaviour, composed of known deterministic functions $g_{ij}$ (with a common choice being low-order polynomials) of the active variables $x_{A_i}$, which are a subset of the inputs that are found to be most influential for output $f_i(x)$, and of the $b_{ij}$ which are unknown regression coefficients. The middle term $u_i(x_{A_i})$ is a Gaussian process with various forms of correlation structure available, capable of mimicking large classes of functions, which has the flexibility to capture more local behaviour of $f_i(x)$, and $v_i(x)$ is an uncorrelated nugget that represents the effect of the remaining inactive input variables, and/or any stochasticity exhibited by the model.

We perform an initial space filling set of $n = 125$ runs $D = (f(x^{(1)}), f(x^{(2)}), \ldots, f(x^{(n)}))$ with the $x^{(i)} \in \mathcal{X}_0$ chosen using a maximin Latin hypercube design. The emulators are updated by the runs $D$ using the Bayes linear update equations [75], and hence can give a prediction with corresponding uncertainty, of the unobserved $f(x)$ at a new, previously unevaluated input point $x$, in the form of the adjusted

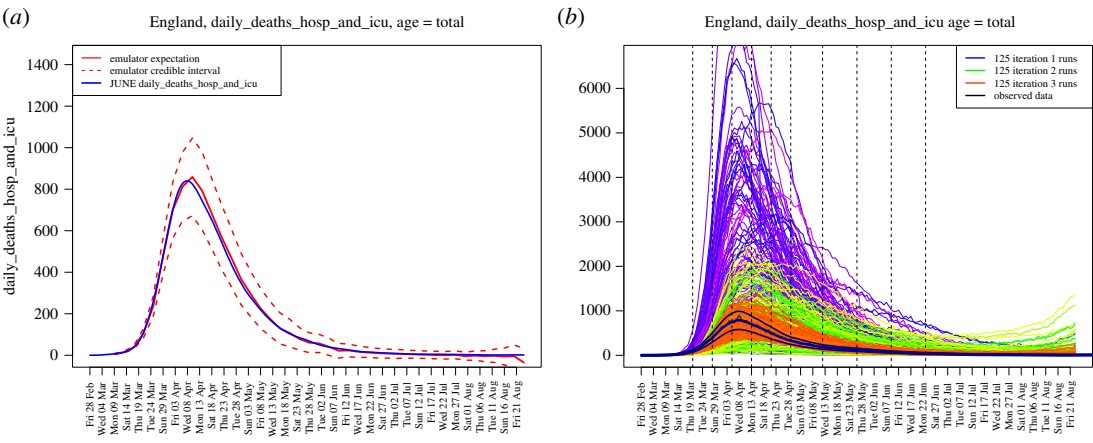

**Figure 28.** (a) An example diagnostic showing the emulator prediction $E_D(f_i(x))$ for $f_i(x)$ across several time points (the solid red line) and the prediction interval $E_D(f_i(x)) \pm 3\sqrt{Var_D(f_i(x))}$ (the red dashed lines) along with the held out smoothed run output $f(x)$ (the blue line). The emulator captures the behaviour of the Ｊｕｎｅ model well. (b) Daily hospital deaths for all of England, showing the progression of the runs from iterations 1, 2 and 3 used in the history matching process (in purple, green and red respectively). Observed data (smoothed and original) in black. Vertical dashed lines: emulated outputs.

expectation $E_D(f_i(x))$ and the adjusted variance $Var_D(f_i(x))$, respectively. The emulators have to satisfy extensive diagnostics [75,98], an illustrative example of which is given in figure 28a, which shows the emulator prediction $E_D(f_i(x))$ for $f_i(x)$ across several time points (the solid red line) and the prediction interval $E_D(f_i(x)) \pm 3\sqrt{Var_D(f_i(x))}$ (the red dashed lines) along with the held out run output $f(x)$ (the blue line) which the emulator has not previously seen, showing excellent agreement between emulator and model. The emulator evaluation takes a fraction of a second, and mimics the Ｊｕｎｅ model well.

By confronting the emulators with the observed data vector $z$ corresponding to the outputs in $f$, and incorporating major sources of uncertainty (e.g. observation error, structural model discrepancy, stochasticity), we can rule out large parts of the input parameter space $\mathcal{X}_0$ as implausible. We do this using an implausibility measure, for which the univariate version $I_i(x)$, is defined for each output as

$$I_i^2(x) = \frac{(E_{D_i}(f_i(x)) - z_i)^2}{Var_D(f_i(x)) + \sigma_{\epsilon_i}^2 + \sigma_{e_i}^2},$$ (E 2)

where $E_D(f_i(x))$ and $Var_D(f_i(x))$ are the emulator expectation and variance as before, $z_i$ is the observed data point corresponding to model output $f_i$, $\sigma_{e_i}^2$ is the variance of the observation error $e_i$ (a random quantity representing the imperfections of the measurement process), and $\sigma_{\epsilon_i}^2$ is the variance of the model discrepancy $\epsilon_i$ (an often-neglected random quantity representing the imperfections of the model [74,75,99]). If $I_i(x)$ is large, it is because the emulator expectation for $f_i(x)$ is very far from the observed data $z_i$, even given all the major sources of uncertainty, and therefore the input parameter $x$ is highly unlikely to yield model output similar to observed data were we to evaluate Ｊｕｎｅ there, and hence $x$ could be discarded from further analysis. A typical cutoff may be $I_i(x) < c$ where $c = 3$.

There are various ways to combine implausibility measures for each of the individual outputs, the simplest being to maximize: $I_M(x) = \max_i I_i(x)$, that is to take the maximum implausibility across all outputs of interest, which is the measure chosen here, although we note that other more nuanced and/or robust versions are available, that capture more of the multivariate behaviour [75].

We now employ iterative history matching [75], a parameter search method that seeks to identify all parts of parameter space that would give rise to acceptable matches between model output and observational data. This proceeds at the $j$th iteration (or wave), by constructing emulators using the current set of runs, removing the implausible parts of the input space to define the new non-implausible region $\mathcal{X}_j = \{x \in \mathcal{X}_0 : I_M(x) < c\}$, designing and performing a new space filling set of runs across the reduced input space $\mathcal{X}_j$ and re-emulating, but now with a more accurate emulator defined only over the reduced region $\mathcal{X}_j$. For further discussion see [75,81,82], but it suffices to note that the iterative nature of history matching is key, as it allows later iteration emulators to become far more accurate as they are only employed over far smaller parts of the input space, and are hence informed by a much higher density of runs.

The observed data for total deaths were obtained from the ONS, while the hospital deaths data is taken from CPNS—the Covid Patient Notification System [72]. For each output corresponding to the element of $f$, the data were first smoothed slightly with a standard kernel smoother, to reduce the day-to-day stochasticity. The observation error and model discrepancy variances for each output were each decomposed into multiplicative and additive components to represent possible systematic biases, in addition to a scaled $\sqrt{n}$ component for the observation error only, to model the noisy count process. For example, we have the decompositions $\sigma_{\epsilon_i}^2 = \alpha_{\text{mult},\epsilon_i}^2 z_i^2 + \gamma_{\text{add},\epsilon_i}^2$, with $\alpha_{\text{mult},\epsilon_i} = 0.06$ and $\gamma_{\text{add},\epsilon_i}^2 = 3/2$, and $\sigma_{e_i}^2 = \alpha_{\text{mult},e_i}^2 z_i^2 + \gamma_{\text{add},e_i}^2 + (\delta_{\text{corr},e_i}\sqrt{z_i})^2$, with $\alpha_{\text{mult},e_i} = 0.06$ and $\gamma_{\text{add},e_i}^2 = 3/2$ and $\delta_{\text{corr},e_i} = 0.25$ governed by the mitigation of the smoothing process.

As described in §8, we performed three waves of the history match with 125 runs each wave, finding that the emulators were of sufficient accuracy after the third wave. Figure 28$b$ shows the progression of the runs from iterations 1, 2 and 3 used in the history matching process (in purple, green and red, respectively) for the daily hospital deaths in England output, with the data (original and smoothed) in black. We can see that the third iteration runs are vastly improved and surround the observed data. These allow accurate emulators to be constructed that can identify the region of input space of interest, which were used to construct figure 22, as discussed in §8.

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
