## [Peer Review File · Royal Society Open Science]

Review History

RSOS-210506.R0 (Original submission)

Review form: Reviewer 1

Is the manuscript scientifically sound in its present form?

Yes

Are the interpretations and conclusions justified by the results?

Yes

Is the language acceptable?

Yes

Do you have any ethical concerns with this paper?

No

Have you any concerns about statistical analyses in this paper?

No

Recommendation?

Accept with minor revision (please list in comments)

Comments to the Author(s)

Comments to the author are attached as a .txt file (see Appendix A) file because all line breaks are removed when I try to enter comments through the web interface.

Review form: Reviewer 2**Is the manuscript scientifically sound in its present form?**

Yes

Are the interpretations and conclusions justified by the results?

Yes

Is the language acceptable?

Yes

Do you have any ethical concerns with this paper?

No

Have you any concerns about statistical analyses in this paper?

No

Recommendation?

Accept as is

Comments to the Author(s)

1. Last line on page 7, the word 'accurately' appears twice
2. Section 3.4 - It wasn't very clear to me whether the information about household location was used at all while assigning workplaces?
3. Section 5.1 - I was a bit confused by what is meant by 'group' here and whether the betas were fitted for each location and group. As in Table 6, it seems that they were fit for each type of location only?
4. (Line numbers as given on the bottom of the page) Page 18 line 54/55 Typo - 'incrasing' -> 'increasing'
5. Page 19 line 53 Typo - 'captures' -> 'captured'
6. Page 20 line 29 Typo - 'oustide' -> 'outside'
7. Page 23 line 52 is incomplete
8. Page 24 line 8 Typo - 'measureing' -> 'measure in a'

Decision letter (RSOS-210506.R0)

Dear Professor Krauss

On behalf of the Editors, we are pleased to inform you that your Manuscript RSOS-210506 "JUNE: open-source individual-based epidemiology simulation" has been accepted for publication in Royal Society Open Science subject to minor revision in accordance with the referees' reports. Please find the referees' comments along with any feedback from the Editors below my signature.

Please submit your revised manuscript and required files (see below) no later than 7 days from today's (ie 09-Jun-2021) date. Note: the ScholarOne system will 'lock' if submission of the revision is attempted 7 or more days after the deadline. If you do not think you will be able to meet this deadline please contact the editorial office immediately.

on behalf of Dr Feng Fu (Associate Editor) and Mark Chaplain (Subject Editor)
openscience@royalsociety.org

Reviewer comments to Author:

Reviewer: 1

Comments to the Author(s)

Comments to the author are attached as a .txt file because all line breaks are removed when I try to enter comments through the web interface.

Reviewer: 2

Comments to the Author(s)

1. Last line on page 7, the word 'accurately' appears twice
2. Section 3.4 - It wasn't very clear to me whether the information about household location was used at all while assigning workplaces?
3. Section 5.1 - I was a bit confused by what is meant by 'group' here and whether the betas were fitted for each location and group. As in Table 6, it seems that they were fit for each type of location only?
4. (Line numbers as given on the bottom of the page) Page 18 line 54/55 Typo - 'incrasing' -> 'increasing'
5. Page 19 line 53 Typo - 'captures' -> 'captured'
6. Page 20 line 29 Typo - 'oustide' -> 'outside'
7. Page 23 line 52 is incomplete

8. Page 24 line 8 Typo - 'measureing' -> 'measure in a'

===PREPARING YOUR MANUSCRIPT===

===PREPARING YOUR REVISION IN SCHOLARONE===

Author's Response to Decision Letter for (RSOS-210506.R0)

See Appendix B.

Decision letter (RSOS-210506.R1)

Dear Professor Krauss,

I am pleased to inform you that your manuscript entitled "JUNE: open-source individual-based epidemiology simulation" is now accepted for publication in Royal Society Open Science.

COVID-19 rapid publication process:

We are taking steps to expedite the publication of research relevant to the pandemic. If you wish, you can opt to have your paper published as soon as it is ready, rather than waiting for it to be published the scheduled Wednesday.

This means your paper will not be included in the weekly media round-up which the Society sends to journalists ahead of publication. However, it will still appear in the COVID-19 Publishing Collection which journalists will be directed to each week (<https://royalsocietypublishing.org/topic/special-collections/novel-coronavirus-outbreak>).

If you wish to have your paper considered for immediate publication, or to discuss further, please notify openscience_proofs@royalsociety.org and press@royalsociety.org when you respond to this email.

You can expect to receive a proof of your article in the near future. Please contact the editorial office (openscience@royalsociety.org) and the production office (openscience_proofs@royalsociety.org) to let us know if you are likely to be away from e-mail contact – if you are going to be away, please nominate a co-author (if available) to manage the proofing process, and ensure they are copied into your email to the journal. Due to rapid publication and an extremely tight schedule, if comments are not received, your paper may experience a delay in publication.

on behalf of Dr Feng Fu (Associate Editor) and Mark Chaplain (Subject Editor)
openscience@royalsociety.org

Appendix A

This manuscript is a nice description of a very complex modeling framework applied to SARS-CoV-2 transmission in England. The open source code and the fully-worked out example of England should be good resource for those wanting to simulate large populations, particularly those similar to England's.

The manuscript is basically a "methods" paper that shows how to build a large synthetic population and to simulate SARS-CoV-2 transmission and behavioral changes. The manuscript has more material in the main body and less in the Appendices than I expected, so it might be hard for a non-specialist to read. But this is not a major concern since results are not the main focus of the paper.

The number of parameters is daunting, and the authors describe their Bayesian history matching approach to fitting 18 free parameters. It would be nice to know if the parameters look "reasonable". For example, do the relative contributions of different venues to COVID transmission seem plausible, or are there any relationship between major variables that can be described? Are there any parameters that could not be estimated? And when you present model results, like Figures 18-20, do you sample over the whole plausible region of parameter space?

More detailed comments follow:

Will you have a version of the model associated with this manuscript? I imagine that JUNE is an evolving project, so it would be nice to be able to associate this detailed description with a version of the code in your repos.

Section 3.2. Why not use census microdata to construct households? Does UK microdata have sample households with the age, sex, and ethnicity of all members?

<https://www.ons.gov.uk/census/2011census/2011censusdata/censusmicrodata>

In Section A.1, I am not clear on how individuals are assigned to households. It seems that you start with a set of individuals with exact ages, genders, and ethnicity, but there is no algorithm described in A.1 to match them to households. Does your algorithm assume all couples are heterosexual (opposite sex couples)? I assume ethnicities are distributed randomly across households in an output area.

Section 3.3. Children age 0 attend school? Are there children who do not attend school (especially those <4y)? What happens to them in the model? Are there daycares or stay-at-home parents/caretakers? In Figure 6a, it looks like children ages 0 through 18y have similar leisure activities.

Be slightly more specific about teachers. What is the "available population"? I assume it is based on age, geography, and people employed in the "Education" sector?

Section 4.1. Can you describe the actual time steps used in your COVID model? It sounds like there are different weekday and weekend time steps. Does the whole population progresses through time using the same time

steps, or can each person have different time steps? Does transmission occur at night (presumably in the household)?

Section 4.2. Where do the restaurant, pub, and store locations come from?

Figure 8b. I can't see areas with a high number of external commuters (maybe due to the resolution of the map?) If all areas far from London have 0 commuters, maybe zooming in on the area surrounding London would be helpful.

Section 5.1. I don't understand the incubation period distribution - the text says it is centered at two days prior onset of symptoms, which does not sound right. And how do you get a time-dependent infectiousness profile for asymptomatic people, where there is no "incubation period" (because there are no symptoms)?

Figure 10. Caption describes 3 curves (severe, mild, asymptomatic), but figure has 2 curves.

Do large households have the same amount of transmission as small households? That is, if there is a sick person in a household with two people, does that person transmit to the same number of household members as a sick person in a large household or in a dormitory? I'm assuming this based on Equation 5.1.

Figure 9. The color bars have no units. Is this all relative and scaled to 0.0-1.0?

Section 5.2, Figure 11. Can patients bounce between Hospitalised and Intensive Care indefinitely? Trajectories listed in Table 3 indicate that there is a maximum of one trip to the ICU.

It would be good to define "case". Is it anyone who is infected (including asymptomatics)?

Section 5.3. Don't regions in England "seed" each other? I am unclear how much seeding needs to be from "outside" the model. And Equation 5.14 seems to say that you seed all daily cases to a region until the epidemic reaches 1/10 of its maximum. After introducing the first couple of cases, don't they transmit enough to sustain epidemic growth?

Figure 12. In the caption, specify what the black vertical interval is (Looks like the other frameworks have a 95% CI but not JUNE? Are the JUNE results averaged over several runs or are they the result of one realization?)

Figure 13. Are these rates of outcomes or are they stacked histograms (proportions of outcomes)?

Section 6.1. The sentence describing "shielding" of the elderly looks incomplete.

Figure 15. Beta is reduced by the same amount in pubs, schools, and hospitals? This seems non-intuitive. Does this mean that pub customers stay masked and distanced from each other?

Section 6.2. Making people stay completely at home when their workplaces and pubs are closed seems extreme. When schools close, do children have any contacts outside the household? Is there any compensation with other behaviors when activities are forbidden or venues are closed?

Section 6.3. What is the source for Table 4? The table could indicate which items are implemented in the model.

Figure 18. Could you add the number of model realizations to the caption? And are the colors arbitrary? And if you

Section 8. Can you determine how much COVID transmission is from households, schools, workplaces, pubs, etc? This would also have a major impact on the effectiveness of closing different kinds of venues. Did the model-fitting indicate tight ranges for any of the betas? It seems like many of these wouldn't be identifiable, but reporting which could be estimated with some confidence would be useful. The emulation time is stated here. It would be nice to know how much time a full run of JUNE takes. And does the JUNE model of England run on a single processor?

Figure 21. I assume light color is more plausible. Are there meaningful values, like an implausibility cutoff, that can be depicted with the color scale? And should there be 18 x 18 panels? I see 12x12. Some of the parameter names are cut off and hard to read, and it is not clear what the values below the parameter name mean. And a minor point: "June" needs to be "JUNE" in the caption.

Section A.2, Figure 24. The caption indicates one of the plots is distance traveled, but all of the x-axes say school size.

Section A.4 Do people in the same carriages have the same home/work locations? How do you estimate commute time? (Asking because of Figure 6b and to better understand time steps in the model.)

In Appendix D, it is not clear how the combined implausibility measure is defined. Is it the maximum implausibility across outputs?

I don't think that "alpha_seedstrength" or "M_sd4randomfactorall" parameters are defined.

When choosing dates for the emulators to fit, does using dates after schools re-open in June help estimate beta_school? Would adding dates in September and October help estimate beta_university?

In Figure 27, are the model predictions outside the 3stdev window starting in August? Did JUNE capture the behavior of all regions in England and age bins well?

Appendix B

Prof Frank Krauss
Institute for Data Science
Institute for Particle Physics Phenomenology
Department of Physics
Durham University
Durham DH1 3LE

Tel: +44 191 3343751

email: frank.krauss@durham.ac.uk

Dear Editor and Reviewers

we would like to resubmit the revised version of our paper titled “JUNE: open-source individual-based epidemiology simulation” for publication in Royal Society Open Science. We would also like to use the opportunity to thank the reviewers for finding time to carefully read our paper and make suggestions which will most certainly improve its quality. We followed most of the suggestions - please find below a more detailed list of our answers and changes. As requested by the journal, we also submit two versions of the revised manuscript, with and without highlighting the changes and improvements.

All the Best

Frank Krauss (for the authors)

Reviewer 1:

1. For example, do the relative contributions of different venues to COVID transmission seem plausible, or are there any relationship between major variables that can be described? Are there any parameters that could not be estimated? And when you present model results, like Figures 18-20, do you sample over the whole plausible region of parameter space?

Locations of infections across the 14 realizations illustrated in the manuscript demonstrate a consistent hierarchy of locations (an example with typical values has been added in the new Figure 21). In all realisations, households, schools, and companies dominate where people get infected, showing plausible agreement with reality.

6 parameters out of the 18: `beta_cinema`, `beta_university`, `beta_intercity_transport`, `beta_hospital`, `beta_household_visits`, and `alpha_physical`, could not be currently estimated as they had subdominant effects on the model output (this is picked up in the emulator construction process). A comment has been added in section 8 to highlight this.

The model results presented here are from a small example set of runs uniformly sampled over the non-implausible region. There is a comment at the beginning of section 7 to clarify this. Although there is much to discuss regarding the calibration results, due to the scope of this paper, we would wish to defer further analysis of the calibration to future work.

Institute for Particle Physics Phenomenology

Ogden Centre for Fundamental Physics Department of Physics University of Durham
Durham DH1 3LE United Kingdom

Tel: +44-(0)191-334-3811 **Fax:** +44-(0)191-334-3658

2. Will you have a version of the model associated with this manuscript? I imagine that JUNE is an evolving project, so it would be nice to be able to associate this detailed description with a version of the code in your repos.

V1.0 is tagged in GitHub as the release corresponding to the current paper, and is also frozen in Zenodo <https://zenodo.org/record/4925939>

3. Section 3.2. Why not use census micro-data to construct households? Do UK micro-data have sample households with the age, sex, and ethnicity of all members?
<https://www.ons.gov.uk/census/2011census/2011censusdata/censusmicrodata>

The datasets we are using (KS1015EW, KS405UK, DC1104EW), cited in the paper, are already the most detailed available at the Output Area level – typically units of 250 residents.

4. In Section A.1, I am not clear on how individuals are assigned to households. It seems that you start with a set of individuals with exact ages, genders, and ethnicity, but there is no algorithm described in A.1 to match them to households. Does your algorithm assume all couples are heterosexual (opposite sex couples)? I assume ethnicities are distributed randomly across households in an output area.

At the moment, ethnicity is indeed distributed randomly, as indicated by the referee, and indeed we plan to improve on this in a future version of JUNE. The algorithm as it stands attempts to match couples assuming they are heterosexual. However, in the absence of available matches, this requirement is ignored, and as a consequence not all couples will be heterosexual. There is also age matching as specified in A1 (according to marriage age differences). The idea is that the household types are sampled first, and then the algorithm loops over the household types and finds suitable people in the local area for each type. The way we match the local population to the specific type is detailed in an extended description in A.1.

5. Section 3.3. Children age 0 attend school? Are there children who do not attend school (especially those <4y)? What happens to them in the model? Are there day-cares or stay-at-home parents/caretakers? In Figure 6a, it looks like children ages 0 through 18y have similar leisure activities.

As stated in the text, all children aged 5 to 18 attend school. Children in age groups 0 to 4 and 18 to 19 will only attend school if there are available free positions in a nearby school that accepts their age group, after children aged between 5 and 18 have been allocated. The age range each school can accept is extracted from <https://www.ukrlp.co.uk/> together with the school's geo-coordinates. Children not attending school will stay at home and be possibly unattended if all parents have been assigned work.

6. Be slightly more specific about teachers. What is the "available population"? I assume it is based on age, geography, and people employed in the "Education" sector?

We've added a comment specifying that teachers are determined based on where they live, their age (over 21 so that we assume they have attended university), and if they have already been assigned the "Education" sector.

7. Section 4.1. Can you describe the actual time steps used in your COVID model? It sounds like there are different weekday and weekend time steps. Does the whole population progresses through time using the same time steps, or can each person have different time steps? Does transmission occur at night (presumably in the household)?

We have added a new section to the appendix and referred to it in sec 4.1. The new section also contains two tables which describe time-steps in "real-world" time (ie, commute step listed as 08:00-09:00).

Institute for Particle Physics Phenomenology

Ogden Centre for Fundamental Physics Department of Physics University of Durham
Durham DH1 3LE United Kingdom

Tel: +44-(0)191-334-3811 Fax: +44-(0)191-334-3658

8. Section 4.2. Where do the restaurant, pub, and store locations come from?

We added a reference to Open Street Map.

9. Figure 8b. I can't see areas with a high number of external commuters (maybe due to the resolution of the map?) If all areas far from London have 0 commuters, maybe zooming in on the area surrounding London would be helpful.

Any areas which can be seen on the map (which aren't pure white) have >0 external commuters. There is a visible red ring around London which shows where most of the high numbers of commuting are. The purpose of the map is to show that there is a significant, non-zero, number of people commuting into London from all over the UK which is important for inter-regional transmission. This has been clarified in the caption and is highlighted in the text.

10. Section 5.1. I don't understand the incubation period distribution - the text says it is centered at two days prior onset of symptoms, which does not sound right. And how do you get a time-dependent infectiousness profile for asymptomatic people, where there is no "incubation period" (because there are no symptoms)?

We followed reference [32], that determines that maximum infectiousness happens on average about two days before symptoms onset. Note that the distribution is not symmetrical. Regarding asymptomatic people, the incubation period is defined for every infected person regardless of their future symptom trajectory. No one shows symptoms during the incubation period. We further clarified this in the text.

11. Figure 10. Caption describes 3 curves (severe, mild, asymptomatic), but figure has 2 curves.

We fixed the caption – thanks for spotting this inconsistency!

12. Do large households have the same amount of transmission as small households? That is, if there is a sick person in a household with two people, does that person transmit to the same number of household members as a sick person in a large household or in a dormitory? I'm assuming this based on Equation 5.1.

In a two person household, the infected individual “spends” all their contacts on the other person, so it will be very likely to infect the susceptible person. In a larger household, the person does the same number of contacts but distributed among more people (this is implicit in the $1/N_g$ factor in eq. 5.1, and we do not think we should add any further discussion of it). So every other individual in the household will be less likely to be infected (compared to the two person household case). However, given how much time is spent in households, it is highly likely that other household members will become infected over the course of several days and therefore larger households will pose a greater risk to infecting more people.

13. Figure 9. The color bars have no units. Is this all relative and scaled to 0.0-1.0?

We've added an explanation: “Colour bars show (average) number of contacts in social settings between age groups, with all colour scales truncated at one to show differences between settings, while still clearly showing the structure in the matrices.”

14. Section 5.2, Figure 11. Can patients bounce between Hospitalised and Intensive Care indefinitely? Trajectories listed in Table 3 indicate that there is a maximum of one trip to the ICU.

There is indeed only one trip to the ICU, we have added a clarification in the caption.

Institute for Particle Physics Phenomenology

Ogden Centre for Fundamental Physics Department of Physics University of Durham
Durham DH1 3LE United Kingdom

Tel: +44-(0)191-334-3811 Fax: +44-(0)191-334-3658

15. It would be good to define "case". Is it anyone who is infected (including asymptomatics)?

The use of the term is dependent on whether we are discussing cases in the model or cases in reality. We have added a paragraph clarifying this at the beginning of Section 5.

16. Section 5.3. Don't regions in England "seed" each other? I am unclear how much seeding needs to be from "outside" the model. And Equation 5.14 seems to say that you seed all daily cases to a region until the epidemic reaches 1/10 of its maximum. After introducing the first couple of cases, don't they transmit enough to sustain epidemic growth?

Given that we carefully replicate the transport and mixing dynamics within and between regions (note that regions has a specific meaning in JUNE as a collection of MSOAs) then in theory pure case zero importation events could be initiated and cross seeding of regions allowed to run. However, we have good data on the MSOAs for both hospitalisations and mortality early in the and we use them to set up the initial model conditions.

As noted the total number of infections in the community is a latent variable and is inferred from mortality data and later community transmission surveys. The data on the starting point of the spread of the SARS-CoV-2 is based on hospitalizations and infections in February and March. Hence the starting seed conditions are driven by cases and interactions that are presumed to have generated the set of initial observations. Data allows us then to populate a sufficient number of cases within an NHS trust region and these are distributed into the local community in clusters. The 0.1 in Equation 5.14 does refer to the progression, in the sense that we have a cumulative rate of infection from the number of cases presumed to be needed to generate the number of deaths later observed at the peak. Hence 0.1 should be interpreted as the seeding timing to generate the first 10% of cases observed in a local area judged by the peak of deaths. This is noted on page 22 under Equation 5.14.

17. Figure 12. In the caption, specify what the black vertical interval is (Looks like the other frameworks have a 95% CI but not JUNE? Are the JUNE results averaged over several runs or are they the result of one realization?)

The black vertical lines are the confidence interval from the estimates of infection rates from the REACT community study – we added a corresponding comment to the caption. It is important to note that the infection fatality rates are an input to the model and not an output, therefore they are not the result of a run but an estimate done from data on deaths and cases in England. In figure 12 we compare the estimated Infection Fatality Rates that we use as an input in JUNE to two different publications that estimated them from data.

18. Figure 13. Are these rates of outcomes or are they stacked histograms (proportions of outcomes)?

They are stacked histograms.

19. Section 6.1. The sentence describing "shielding" of the elderly looks incomplete.

We fixed the relevant sentences – thanks for spotting this “interesting” use of grammar.

20. Figure 15. Beta is reduced by the same amount in pubs, schools, and hospitals? This seems non-intuitive. Does this mean that pub customers stay masked and distanced from each other?

We corrected a misleading error in the legend to move hospitals to the household line (i.e. keep the beta in hospitals fixed throughout). The logic for keeping the pubs and cinemas changing at the same rate is that masks are worn in all when they're open and distancing is maintained. When schools fully open in September, these lines become decoupled, however, in this paper we're not focusing on the beta fitting exercises we perform at that time. In addition, many of these beta changes are fitted in reality (as mentioned in the Section) and this figure represents an example of how the betas might change in a given

Institute for Particle Physics Phenomenology

Ogden Centre for Fundamental Physics Department of Physics University of Durham
Durham DH1 3LE United Kingdom

Tel: +44-(0)191-334-3811 Fax: +44-(0)191-334-3658

run.

21. Section 6.2. Making people stay completely at home when their workplaces and pubs are closed seems extreme. When schools close, do children have any contacts outside the household? Is there any compensation with other behaviors when activities are forbidden or venues are closed?

The original wording was incorrect - when agents don't go to work because their workplace is closed down then they are still able to choose to go to a leisure activity (if any are available) or to stay at home. This has been corrected.

22. Section 6.3. What is the source for Table 4? The table could indicate which items are implemented in the model.

Policies implemented in the model have now been highlighted in the table. There are multiple sources for each element in the table, including news and government webpages. Given the rapid changes in government pages, it is not always trivial to find pages which refer to previous policies. We therefore merely listed the policies.

23. Figure 18. Could you add the number of model realizations to the caption? And are the colors arbitrary?

There are 14 realisations plotted in Figures 18 and 19. Colours are arbitrary and are only supposed to separate the runs visually. Caption in Figure 18 has been changed to reflect this.

24. Section 8. Can you determine how much COVID transmission is from households, schools, workplaces, pubs, etc? This would also have a major impact on the effectiveness of closing different kinds of venues. Did the model-fitting indicate tight ranges for any of the betas? It seems like many of these wouldn't be identifiable, but reporting which could be estimated with some confidence would be useful. The emulation time is stated here. It would be nice to know how much time a full run of JUNE takes. And does the JUNE model of England run on a single processor?

A new figure has been included in the results section (Fig. 21) showing locations of infections for a single run. A typical run of JUNE (like those shown in Figure 18) takes 8-10 hours to complete on 64 cores and 128GB of memory. A comment about this has been added to Section 8. Of course, this is a function of how many infections take place, meaning the summer of 2020 runs relatively quickly compared to the beginning of the pandemic, or the second wave in the winter of 2020. Whilst it is possible for an England model of JUNE to be run on a single processor, it would take a long amount of time. JUNE is optimised to be run with multiple cores. The history matching/calibration procedure indicated joint constraints on combinations of the input parameters, for example, there is a clear tradeoff required between `beta_school` and `beta_household`, and similar (but more complex) tradeoffs are identified between several other parameters e.g. `beta_company` and `beta_houshold`; `beta_grocery` and `beta_citytransport`, and `beta_household` and `M_quarantine_household_compliance`. Most parameters were not individually identifiable however, `beta_company` and `beta_carevisits` were reasonably well constrained. Comments summarising the above have been added to the second last paragraph in Sec. 8.

25. Figure 21. I assume light color is more plausible. Are there meaningful values, like an implausibility cutoff, that can be depicted with the color scale? And should there be 18 x 18 panels? I see 12x12. Some of the parameter names are cut off and hard to read, and it is not clear what the values below the parameter name mean. And a minor point: "June" needs to be "JUNE" in the caption.

The plot has been improved so parameter names are no longer cut off; a comment added in the caption re the range values below the parameter names, and "June" has been changed to "JUNE". Regarding only 12 inputs being shown: a comment has been added to the caption stating that the 12 most interesting inputs are shown, and a similar comment in the text of section 8. In this plot, the colours just denote the optical depth or thickness of the non-implausible region, which give an idea of where the majority of

Institute for Particle Physics Phenomenology

Ogden Centre for Fundamental Physics Department of Physics University of Durham
Durham DH1 3LE United Kingdom

Tel: +44-(0)191-334-3811 Fax: +44-(0)191-334-3658

non-implausible solutions lie.

26. Section A.2, Figure 24. The caption indicates one of the plots is distance traveled, but all of the x-axes say school size.

This erroneous caption has been fixed.

27. Section A.4 Do people in the same carriages have the same home/work locations? How do you estimate commute time? (Asking because of Figure 6b and to better understand time steps in the model.)

The gateway station chosen by agents is based on the proximity to where they live - i.e. the gateway stations draw out Voronoi polygons around the city from where they draw people from. In the simulations we ran, there are only two commuting time steps in each day - one in the morning and one in the evening. The carriages consist of a random subset of people drawn from all possible people travelling through that gateway station to simulate mixing with different people from different regions who get onto your form of transport at different parts of the journey. We have slightly edited the algorithm in A.4 to better reflect this, and the new time step appendix (B) should clarify this further.

28. In Appendix D, it is not clear how the combined implausibility measure is defined. Is it the maximum implausibility across outputs?

Yes, we choose the maximum implausibility across all outputs. An extra comment has been added to Appendix E to clarify.

29. I don't think that "alpha_seedstrength" or "M_sd4randomfactorall" parameters are defined.

We have added their definition in the caption of Table 8 (Appendix E). Thanks for spotting this!

30. When choosing dates for the emulators to fit, does using dates after schools re-open in June help estimate beta_school? Would adding dates in September and October help estimate beta_university?

A good question: the effect of including outputs in the month of June is to mildly disfavour high values of beta_school, although as can be seen, high values are still possible/required depending upon the values of other parameters (e.g. if beta_household is low). Adding dates in Sept/Oct could help with the estimation of beta_university: current ongoing calibration efforts show a mild affect of beta_university on the outputs mainly in Oct/Nov, however this is subdominant to several other stronger input parameters so we might not learn much directly about beta_university (but learn about it jointly with other parameters). There is much more to be said about the results of such an uncertainty analysis/calibration, however, due to the length and scope of this current paper we would wish to defer detailed discussions to future work.

31. In Figure 27, are the model predictions outside the 3std dev window starting in August? Did JUNE capture the behavior of all regions in England and age bins well?

Running the cluster of model predictions from the non-implausible region forward past August generally leads to a spread of runs that encapsulates the observed data (implying that we can calibrate further using such Autumn data), however this spread does depend on any other summer/autumn parameters we may wish to include, and on how tightly we may wish to match to the first wave data. Again, although interesting, this takes us somewhat beyond the remit of this current paper. JUNE's ability to match all regions and all age bins, as evidenced by Figures 18 and 19, was modestly successful.

Reviewer 2:

1. Last line on page 7, the word 'accurately' appears twice [section 3.2: communal households].

Institute for Particle Physics Phenomenology

Ogden Centre for Fundamental Physics Department of Physics University of Durham
Durham DH1 3LE United Kingdom

Tel: +44-(0)191-334-3811 Fax: +44-(0)191-334-3658

We fixed the sentence.

2. Section 3.4 – It wasn't very clear to me whether the information about household location was used at all while assigning workplaces?

People are assigned a place to live based on the census and we then use other ONS data which tells us what sectors people work in based on where they live. Another dataset informs where people work vs. where they live which allows us to construct an origin-destination matrix which informs the distribution of people to companies based on their living and working locations, and matching the sectors. We have added an emphasis on how sector distribution is a function of residential location which now reads: "In a next step, individuals who are eligible to work (i.e. between the ages of 18-65) are assigned an industry sector based on the geographic distribution of where the workforce live by sector".

3. Section 5.1 – I was a bit confused by what is meant by 'group' here and whether the betas were fitted for each location and group. As in Table 6, it seems that they were fit for each type of location only?

We have clarified this in a footnote in the section. The location, L , refers to different types of location where as the groups, g , refer to the group of people in that location at a given point in time. Therefore across different locations of the same type (e.g. two pubs in different regions), the value N_g (the number of people in each group) will be different but in reality, as you point out, the beta value is fitted per location type, L , and therefore is shared between the two.

Without going into detail we also fixed the typos.

Institute for Particle Physics Phenomenology

Ogden Centre for Fundamental Physics Department of Physics University of Durham
Durham DH1 3LE United Kingdom

Tel: +44-(0)191-334-3811 Fax: +44-(0)191-334-3658